# UNIFIED GENERATIVE MODELING OF 3D MOLECULES VIA BAYESIAN FLOW NETWORKS

**Yuxuan Song**[1*]**, Jingjing Gong**[1*]**, Hao Zhou**[1]**, Mingyue Zheng**[2]**, Jingjing Liu**[1] **& Wei-Ying Ma**[1]

[1] Institute of AI Industry Research (AIR), Tsinghua University
[2] Shanghai Institute of Materia Medica, Chinese Academy of Sciences
{songyuxuan,gongjingjing,zhouhao,maweiying}@air.tsinghua.edu

## ABSTRACT

Advanced generative model (*e.g.*, diffusion model) derived from simplified continuity assumptions of data distribution, though showing promising progress, has been difficult to apply directly to geometry generation applications due to the *multimodality* and *noise-sensitive* nature of molecule geometry. This work introduces Geometric Bayesian Flow Networks (GeoBFN), which naturally fits molecule geometry by modeling diverse modalities in the differentiable parameter space of distributions. GeoBFN maintains the SE-(3) invariant density modeling property by incorporating equivariant inter-dependency modeling on parameters of distributions and unifying the probabilistic modeling of different modalities. Through optimized training and sampling techniques, we demonstrate that GeoBFN achieves state-of-the-art performance on multiple 3D molecule generation benchmarks in terms of generation quality (90.87% molecule stability in QM9 and 85.6% atom stability in GEOM-DRUG[1]). GeoBFN can also conduct sampling with any number of steps to reach an optimal trade-off between efficiency and quality (*e.g.*, 20× speedup without sacrificing performance).

## 1 INTRODUCTION

Molecular geometries can be represented as three-dimensional point clouds, characterized by their Cartesian coordinates in space and enriched with descriptive features. For example, proteins can be represented as proximity spatial graphs (Jing et al., 2021) and molecules as atomic graphs in 3D (Schütt et al., 2017). Thus, learning geometric generative models has the potential to benefit scientific discoveries such as material and drug design. Recent progress in deep generative modeling has paved the way for geometric generative modeling. For example, Gebauer et al. (2019); Luo & Ji (2021) and Satorras et al. (2021a) use autoregressive models and flow-based models, respectively, for generating 3D molecules in-silico. Most recently, inspired by the huge success of diffusion model (DM) in image generation Meng et al. (2022); Ho et al. (2020)

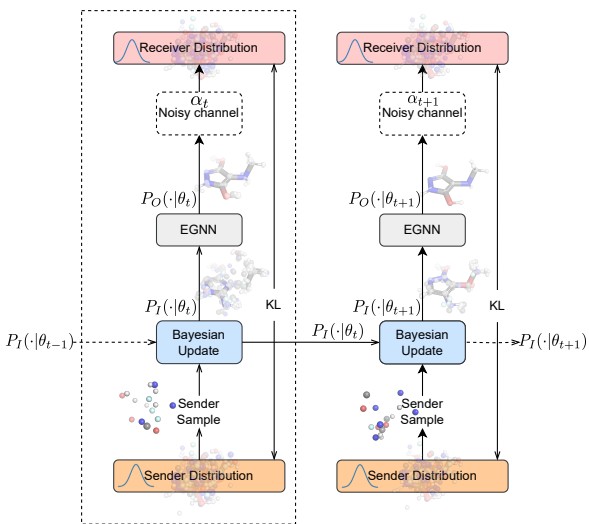

Figure 1: The framework of GeoBFN

and beyond Li et al. (2022), DM incorporating geometric symmetries has been widely explored in the field of geometry generation Hoogeboom et al. (2022); Xu et al. (2023).

---

*Equal Contribution. Correspondence to Hao Zhou(zhouhao@air.tsinghua.edu).

[1]The scores are reported at 1k sampling steps for fair comparison, and our scores could be further improved if sampling sufficiently longer steps

However, two major challenges remain in directly applying DM to molecule geometry: *multi-modality* and *noise sensitivity*. The *multi-modality* issue refers to the dependency on diverse data forms to effectively depict the atomic-level geometry of a molecule. For instance, the continuous variable of atom coordinates is essential for describing the spatial arrangement, while either the discretised atom charge or categorical atom types are employed to completely determine the molecule's composition. *Noise sensitivity* refers to the fact that applying noise or perturbing the atom coordinates will not only change the value of the variable but also have a significant impact on the relationship among different atoms as the Euclidean distances are also changed. Therefore, a small noise on atom coordinates could bring a sudden drop of the signal at the molecule level.

To alleviate these issues, Xu et al. (2023) introduces a latent space for alleviating the inconsistency of unified Gaussian diffusion on different modalities. Anand & Achim (2022) propose to use decomposed modeling of different modalities. Peng et al. (2023) use different noise schedulers for different modalities to accommodate noise sensitivity. However, these methods either depend on the sophisticated and artifact-filled design or lack of guarantee or constraint on the designed space.

In this work, we propose Geometric Bayesian Flow Networks (GeoBFN) to model 3D molecule geometry in a principally different way. Bayesian Flow Networks Graves et al. (2023) (BFN) is a novel generative model developed quite recently. Taking a unique approach by incorporating Bayesian inference to modify the parameters of a collection of independent distributions, brings a fresh perspective to geometric generative modeling. Firstly, GeoBFN uses a unified probabilistic modeling formulation for different modalities in the molecule geometry. Secondly, regarding the variable of 3D atom coordinates, the input variance for BFNs is considerably lower than DMs, leading to better compatibility with the inherent *noise sensitivity*. Further, we bring the geometry symmetries into the Bayesian update procedure through an equivariant inter-dependency modeling module. We also demonstrate that the density function of implied generative distribution is SE-(3) invariant and the generative process of iterative updating is roto-translational equivariant. Thirdly, with BFN's powerful probabilistic modeling capacity, 3D molecule geometry representation can be further optimized into a representation with only two similar modalities: discretised charge and continuous atom coordinates. The mode-redundancy issue on discretised variable in the original BFNs is fixed by an early mode-seeking sampling strategy in GeoBFN.

With operating on the space with less variance, GeoBFN could sample with any number of steps which provides a superior trade-off between efficiency and quality, which leads to a $20\times$ speedup with competitive performance. Besides, GeoBFN is a general framework that can be easily extended to other molecular tasks. We conduct thorough evaluations of GeoBFN on multiple benchmarks, including both unconditional and property-conditioned molecule generation tasks. Results demonstrate that GeoBFN consistently achieves state-of-the-art generation performance on molecule stability and other metrics. Empirical studies also show a significant improvement in controllable generation and demonstrate that GeoBFN enjoys a significantly higher modeling capacity and inference efficiency.

## 2 PRELIMINARIES

### 2.1 SE-(3) INVARIANT DENSITY MODELING

To distinguish geometry representation and the atomic property features, we use the tuple $\mathbf{g} = \langle \mathbf{x}, \mathbf{h} \rangle$ to represent the 3D molecules. Note here $\mathbf{x} = (\mathbf{x}^1, \ldots, \mathbf{x}^N) \in \mathbb{R}^{N \times 3}$ is the atom coordinate matrix, and $\mathbf{h} = (\mathbf{h}^1, \ldots, \mathbf{h}^N) \in \mathbb{R}^{N \times d}$ is the node feature matrix, *e.g.*, atomic types and charges. Density estimation on the 3D molecules should satisfy specific symmetry conditions of the geometry. In this work, we focus on the transformations $T_g$ in the Special Euclidean group (SE-(3)), *i.e.*, the group of rotation and translation in 3D space, where transformations $T_g$ can be represented by a translation $\mathbf{t}$ and an orthogonal matrix rotation $\mathbf{R}$. Note for a generative model on molecule geometry with underlying density function $p_{\boldsymbol{\theta}}(\langle \mathbf{x}, \mathbf{h} \rangle)$, the likelihood should not be influenced by the rotation or translation of the entire molecule, which means the likelihood function should be SE-(3) invariant on the input coordinates, *i.e.*, $p_{\boldsymbol{\theta}}(\langle \mathbf{x}, \mathbf{h} \rangle) = p_{\boldsymbol{\theta}}(\langle \mathbf{R}\mathbf{x} + \mathbf{t}, \mathbf{h} \rangle)$.

### 2.2 BAYESIAN FLOW NETWORKS

The Bayesian Flow Networks (BFNs) are based on the following latent variable models: for learning the probability distribution $p_{\boldsymbol{\theta}}$ over $\mathbf{g}$, a series of noisy versions $\langle \mathbf{y}_1, \cdots, \mathbf{y}_n \rangle$ of $\mathbf{g}$ are introduced as

latent variables. And then the *variational lower bound* of likelihood is optimized:

$$\log p_{\boldsymbol{\theta}}(\mathbf{g}) \geq \mathop{\mathbb{E}}_{\mathbf{y}_1,\ldots,\mathbf{y}_n \sim q} \left[ \log \frac{p_{\phi}(\mathbf{g} \mid \mathbf{y}_1,\ldots,\mathbf{y}_n) p_{\phi}(\mathbf{y}_1,\ldots,\mathbf{y}_n)}{q(\mathbf{y}_1,\ldots,\mathbf{y}_n|\mathbf{g})} \right]$$

$$= -D_{KL}(q\|p_{\phi}(\mathbf{y}_1,\ldots,\mathbf{y}_n)) + \mathop{\mathbb{E}}_{\mathbf{y}_1,\ldots,\mathbf{y}_n \sim q} \log [p_{\phi}(\mathbf{g} \mid \mathbf{y}_1,\ldots,\mathbf{y}_n)] \quad (1)$$

And $q$ is namely the variational distribution. The prior distribution of latent variables is usually organized autoregressively, *i.e.*, $p_{\phi}(\mathbf{y}_1,\cdots,\mathbf{y}_n) = p_{\phi}(\mathbf{y}_1)p_{\phi}(\mathbf{y}_2 \mid \mathbf{y}_1)p_{\phi}(\mathbf{y}_n \mid \mathbf{y}_{n-1}\cdots\mathbf{y}_1)$ which also implies the data generation procedure, *i.e.*, $\mathbf{y}_1 \to \cdots \to \mathbf{y}_n \to \mathbf{g}$ (*Note:* this procedure only demonstrates the generation order, yet does NOT imply Markov property for the following derivative).

One widely adopted intuition for the generation process is that the information of the data samples should progressively increase along with the above Markov chain, *e.g.*, noisier images to cleaner images. The key motivation of BFNs is that the information along the latent variables should change as smoothly as possible for all modalities including discretized and discrete variables. To this end, BFNs operate on the distributions in the parameter space, in contrast to the sample space.

We introduce components of BFNs one by one (Fig.2a). Firstly, the variational distribution $q$ is defined by the following form:

$$q(\mathbf{y}_1,\ldots,\mathbf{y}_n \mid \mathbf{g}) = \prod_{i=1}^{n} p_S(\mathbf{y}_i \mid \mathbf{g}; \alpha_i) \quad (2)$$

$p_S(\mathbf{y}_i \mid \mathbf{g}; \alpha_i)$ is termed as the *sender distribution*, which could be seen as adding noise to the data according to a predefined accuracy $\alpha_i$.

Secondly, for the definition of $p_{\phi}$, BFNs will first transfer the noisy sample $\mathbf{y}$ to the parameter space, obtaining $\boldsymbol{\theta}$, then apply Bayesian update in the parameters space and transfer back to the noisy sample space at last. To clarify, $\boldsymbol{\theta}$ refers to the parameter of distributions in the sample space, *e.g.*, the mean/variance for Gaussian distribution or probabilities for categorical distribution. In the scope of BFNs, the distributions on the sample space are factorized by default, *e.g.*, $p(\mathbf{g} \mid \boldsymbol{\theta}) = \prod_{d=1}^{D} p(g^{(d)} \mid \theta^{(d)})$.

Thirdly, a neural network $\Phi$ takes $\boldsymbol{\theta}$ as input and aims to model the dependency among different dimensions hence to recover the distribution of the original sample $\mathbf{g}$. The output of neural network $\Phi(\boldsymbol{\theta})$ still lies in the parameter space, and we termed it as the parameter of *output distribution* $p_O$, where $p_O(\mathbf{y}|\boldsymbol{\theta}; \phi) = \prod_{d=1}^{D} p_O(\mathbf{y}^{(d)} \mid \Phi(\boldsymbol{\theta})^{(d)})$.

To map the noisy sample $\mathbf{y}$ to the input space, Bayesian update is applied to $\boldsymbol{\theta}$:

$$\boldsymbol{\theta}_i \leftarrow h(\boldsymbol{\theta}_{i-1}, \mathbf{y}_i, \alpha_i), \quad (3)$$

$h$ is called *Bayesian update function* . The distribution over $(\boldsymbol{\theta}_0, \ldots, \boldsymbol{\theta}_{n-1})$ is then defined by the *Bayesian update distribution* via marginalizing out $\mathbf{y}$:

$$p_{\phi}(\boldsymbol{\theta}_0,\ldots,\boldsymbol{\theta}_{n-1}) = p(\boldsymbol{\theta}_0) \prod_{i=1}^{n} p_U(\boldsymbol{\theta}_i \mid \boldsymbol{\theta}_{i-1}; \alpha_i), \quad (4)$$

where $p(\boldsymbol{\theta}_0)$ is a simple prior for ease of generation, *e.g.*, standard normal, and $p_U$ could be obtained from Eq. 3:

$$p_U(\boldsymbol{\theta}_i \mid \boldsymbol{\theta}_{i-1}; \alpha_i) = \mathop{\mathbb{E}}_{p_R(\mathbf{y}_i|\boldsymbol{\theta}_{i-1};\alpha_i)} \delta(\boldsymbol{\theta}_i - h(\boldsymbol{\theta}_{i-1}, \mathbf{y}_i, \alpha_i)), \quad (5)$$

$\delta$ being the Dirac delta distribution. $p_R(\mathbf{y}_i|\boldsymbol{\theta}_{i-1}, \alpha_i) = \mathop{\mathbb{E}}_{p_O(\mathbf{x}'|\boldsymbol{\theta}_{i-1};\phi)} p_S(\mathbf{y}_i|\mathbf{x}'; \alpha_i)$ and is also called the as *receiver distribution*.

At last we map $\Phi(\boldsymbol{\theta})$ back to the noisy sample space by combining the known form, accuracy of $P_S$ and marginalizing out $\mathbf{y}$:

$$p_{\phi}(\mathbf{y}_1,\ldots,\mathbf{y}_n) = p_{\phi}(\mathbf{y}_1) \prod_{i=2}^{n} p_{\phi}(\mathbf{y}_i \mid \mathbf{y}_{\{1:i-1\}}) = \prod_{i=1}^{n} p_{\phi}(\mathbf{y}_i \mid \boldsymbol{\theta}_{i-1})$$

$$= \prod_{i=1}^{n} \mathop{\mathbb{E}}_{p_O(\mathbf{x}'_i|\boldsymbol{\theta}_{i-1};\phi)} [p_S(\mathbf{y}_i|\mathbf{x}'_i; \alpha_i)], \quad (6)$$

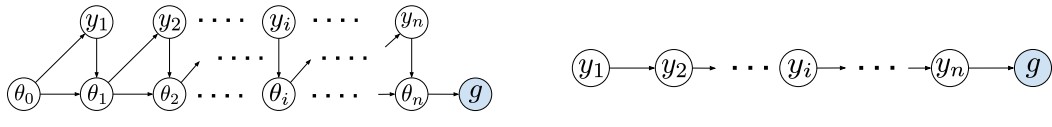

Figure 2: Graphical View of Comparison between BFN and Diffusion

where we use $\boldsymbol{\theta}_{0:n-1}$ to abbreviate $(\boldsymbol{\theta}_0, \ldots, \boldsymbol{\theta}_{n-1})$, and $\mathbf{y}$ similar. . Till now, we have defined $q$, $p_\phi(\mathbf{y}_1, \ldots, \mathbf{y}_n)$, and $p_\phi(\mathbf{g} \mid \mathbf{y}_1, \ldots, \mathbf{y}_n)$ is simply $p_O(\mathbf{g} \mid \boldsymbol{\theta}_n)$ on each sample, thus Eq.1 can be estimated.

## 3 METHODOLOGY

### 3.1 SE-(3) INVARIANT GEOMETRY DENSITY MODELING

As discussed in Sec. 2.1, for a generative model on the 3D molecule geometry, it is crucial to hold the SE-(3) invariant conditions. Recall the mathematics formula of the geometries $\boldsymbol{g} = \langle \mathbf{x}, \mathbf{h} \rangle$, we denote the latent variable, *e.g.*, noisy samples, of $\mathbf{g}$ as $\mathbf{y}^g$. We are interested in applying the SE-(3) invariant conditions to the probabilistic model $p_\phi$. To this end, we need to first reformulate the likelihood function:

$$p_\phi(\mathbf{g}) = p_\phi(\langle \mathbf{x}, \mathbf{h} \rangle) = \int_{\mathbf{y}_1^g, \cdots, \mathbf{y}_n^g} p_\phi(\mathbf{g} \mid \mathbf{y}_1^g, \cdots, \mathbf{y}_n^g) p_\phi(\mathbf{y}_1^g, \cdots, \mathbf{y}_n^g) d\mathbf{y}_1^g \ldots d\mathbf{y}_n^g. \qquad (7)$$

With $\boldsymbol{\theta}^x$ and $\mathbf{y}^x$ and all the distributions defined in the same way as above, we focus on the variables corresponds to $\mathbf{x}$ in the geometry $\mathbf{y}^g$. Then we have the following theorem:

**Theorem 3.1.** *(SE-(3) Invariant Condition)*

- *With the $\boldsymbol{\theta}^x$, $\mathbf{y}^x$, $\mathbf{x}$ constrained in the zero Center of Mass(CoM) space (Köhler et al., 2020; Xu et al., 2022), the likelihood function $p_\phi$ is translational invariant.*

- *When the following properties are satisfied, the likelihood function $p_\phi$ is roto-invariant:*

$$p_O\left(\mathbf{x}' \mid \boldsymbol{\theta}_{i-1}^x; \phi\right) = p_O\left(\mathbf{R}(\mathbf{x}') \mid \boldsymbol{R}(\boldsymbol{\theta}_{i-1}^x); \phi\right); p_S\left(\mathbf{y}^x \mid \mathbf{x}'; \alpha\right) = p_S\left(\mathbf{R}(\mathbf{y}^x) \mid \mathbf{R}(\mathbf{x}'); \alpha\right);$$
$$h(\mathbf{R}(\boldsymbol{\theta}_{i-1}^x), \mathbf{R}(\mathbf{y}_i^x), \alpha_i) = \mathbf{R}h(\boldsymbol{\theta}_{i-1}^x, \mathbf{y}_i^x, \alpha_i); p(\mathbf{x}'|\boldsymbol{\theta}_0^x) = p(\mathbf{R}(\mathbf{x}')|\boldsymbol{\theta}_0^x), \ \forall \text{ orthogonal } \mathbf{R}$$

**Proposition 3.2.** *With the condition in Theorem. 3.1 satisfied, the evidence lower bound objective in Eq. 1, i.e.,*

$$\mathcal{L}_{VLB}(\mathbf{x}) = \mathop{\mathbb{E}}_{p_\phi\left(\boldsymbol{\theta}_0^x, \ldots, \boldsymbol{\theta}_n^x\right)} \left[ \sum_{i=1}^{n} D_{KL}(p_S\left(\cdot \mid \mathbf{x}; \alpha_i\right) \| p_R(\cdot \mid \boldsymbol{\theta}_{i-1}^x; \alpha_i)) - \log p_\phi\left(\mathbf{x} \mid \boldsymbol{\theta}_n^x\right) \right], \quad (8)$$

*with the Bayesian update distribution $p_\phi\left(\boldsymbol{\theta}_0^x, \ldots, \boldsymbol{\theta}_{n-1}^x\right) = \prod_{i=1}^{n} p_U\left(\boldsymbol{\theta}_i \mid \boldsymbol{\theta}_{i-1}, \mathbf{x}; \alpha_i\right)$ similar to Eq. 4. And $p_R(\cdot \mid \boldsymbol{\theta}_{i-1}^x; \alpha_i) = \mathop{\mathbb{E}}_{p_O(\mathbf{x}_i' \mid \boldsymbol{\theta}_{i-1}^x; \phi)} [p_S(\mathbf{y}_i | \mathbf{x}_i'; \alpha_i)]$, $p_\phi(\mathbf{x} | \boldsymbol{\theta}_n^x) = p_O(\mathbf{x} | \boldsymbol{\theta}_n^x, \phi)$. Derivation from Eq. 1 to equation 8 is at Appendix C.4. if $p_U\left(\boldsymbol{\theta}_i \mid \boldsymbol{\theta}_{i-1}, \mathbf{x}; \alpha_i\right) = p_U\left(\mathbf{R}\boldsymbol{\theta}_i \mid \mathbf{R}\boldsymbol{\theta}_{i-1}, \mathbf{R}\mathbf{x}; \alpha_i\right)$, then $\mathcal{L}_{VLB}(\mathbf{x})$ is also SE-(3) invariant.*

We leave the formal proof of Theorem. 3.1 and Proposition. 3.2 in Appendix C.

### 3.2 GEOMETRIC BAYESIAN FLOW NETWORKS

Then we introduce the detailed formulation of geometric Bayesian flow networks (GeoBFN) based on the analysis in Sec. 3.1. For describing a 3D molecule geometry $\mathbf{g} = \langle \mathbf{x}, \mathbf{h} \rangle$, various representations can be utilized for the node features. The atom types $\mathbf{h_t}$ and atomic charges $\mathbf{h_c}$ are commonly employed, with the former being discrete (categorical) and the latter being discretized (integer). Together with the continuous variable, *e.g.*, atom coordinates $\mathbf{x}$, the network module in the modeling of the *output distribution* of GeoBFN could be parameterized with an equivariant graph neural network (EGNN) (Satorras et al., 2021b) $\Phi$:

$$\Phi(\mathbf{R}\boldsymbol{\theta}^x + \mathbf{t}, [\boldsymbol{\theta}^{h_t}, \boldsymbol{\theta}^{h_c}]) = [\mathbf{R}\boldsymbol{\theta}^{x'} + \mathbf{t}, \boldsymbol{\theta}^{h'_t}, \boldsymbol{\theta}^{h'_c}], \quad \forall \mathbf{R}, \mathbf{t} \qquad (9)$$

where $\Phi(\boldsymbol{\theta}^x, [\boldsymbol{\theta}^{h_t}, \boldsymbol{\theta}^{h_c}]) = [\boldsymbol{\theta}^{x'}, \boldsymbol{\theta}^{h'_t}, \boldsymbol{\theta}^{h'_c}]$. And then we introduce the necessary components to derive the objective in Eq. 8

**Atom Coordinates $\mathbf{x}$ and Charge $\mathbf{h}_c$:** For the continuous and discretized variables, the *input distribution* is set as the factorized Gaussian distributions, where $\boldsymbol{\theta} \overset{\text{def}}{=} \{\mu, \rho\}$ the parameter of $\mathcal{N}\left(\cdot \mid \mu, \rho^{-1}\mathbf{I}\right)$. For simplicity, we take $\mathbf{x}$ as an example to illustrate the common parts of the two variables. And $\boldsymbol{\theta}_0^x$ is set as $\{\mathbf{0}, 1\}$. The *sender distribution* $p_S$ is also an isotropic Gaussian distribution:

$$p_S(\cdot \mid \mathbf{x}; \alpha\mathbf{I}) = \mathcal{N}\left(\mathbf{x}, \alpha^{-1}\mathbf{I}\right) \tag{10}$$

Given the nice property of isotropic Gaussian (proof given by Graves et al. (2023)), the simple form of *Bayesian update function* could be derived as:

$$h\left(\{\boldsymbol{\mu}_{i-1}, \rho_{i-1}\}, \mathbf{y}, \alpha\right) = \{\boldsymbol{\mu}_i, \rho_i\}, \quad \text{Here} \quad \rho_i = \rho_{i-1} + \alpha, \boldsymbol{\mu}_i = \frac{\boldsymbol{\mu}_{i-1}\rho_{i-1} + \mathbf{y}\alpha}{\rho_i} \tag{11}$$

As shown in Eq. 11, the randomness only exists in $\mu$, and the corresponding *Bayesian update distribution* in Eq. 8 is as:

$$p_U\left(\boldsymbol{\theta}_i \mid \boldsymbol{\theta}_{i-1}, \mathbf{x}; \alpha\right) = \mathcal{N}\left(\boldsymbol{\mu}_i \mid \frac{\alpha\mathbf{x} + \boldsymbol{\mu}_{i-1}\rho_{i-1}}{\rho_i}, \frac{\alpha}{\rho_i^2}\mathbf{I}\right) \tag{12}$$

The above discrete-time Bayesian update could be easily extended to continuous-time, with an accuracy scheduler defined as $\beta(t) = \int_{t'=0}^{t} \alpha(t') \, dt', t \in [0, 1]$. Given the accuracy additive property of $p_U$ (proof given by Graves et al. (2023)), $\underset{p_U(\boldsymbol{\theta}_{i-1} \mid \boldsymbol{\theta}_{i-2}, \mathbf{x}; \alpha_a)}{\mathbb{E}} p_U\left(\boldsymbol{\theta}_i \mid \boldsymbol{\theta}_{i-1}, \mathbf{x}; \alpha_b\right) = p_U\left(\boldsymbol{\theta}_i \mid \boldsymbol{\theta}_{i-2}, \mathbf{x}; \alpha_a + \alpha_b\right)$, the *Bayesian flow distribution* could be obtained as:

$$p_F(\boldsymbol{\theta}^x \mid \mathbf{x}; t) = p_U\left(\boldsymbol{\theta}^x \mid \boldsymbol{\theta}_0, \mathbf{x}; \beta(t)\right) \tag{13}$$

The key difference of atom coordinates $\mathbf{x}$ and charges $\mathbf{h}_c$ lies in the design of the *output distribution*. For continuous variable $\mathbf{x}$, the network module $\Phi$ directly outputs an estimated $\hat{\mathbf{x}} = \Phi(\boldsymbol{\theta}^g, t)$. Hence for timestep $t$, the *output distribution* is

$$p_O\left(\mathbf{x}' \mid \boldsymbol{\theta}^g, t; \phi\right) = \delta(\mathbf{x} - \Phi(\boldsymbol{\theta}^g, t)) \tag{14}$$

While for discretized variable $\mathbf{h}_c$, the network module will output two variables, $\boldsymbol{\mu}_{h_c}$ and $\ln \sigma_{h_c}$ with dimension equivalent to $\mathbf{h}_c$ which implies a distribution $\mathcal{N}(\boldsymbol{\mu}_{h_c}, \sigma_{h_c}^2 \mathbf{I})$. With a $K$-bins discretized variable, the support is split into $K$ buckets with each bucket $k$ centered as $k_c = \frac{2k-1}{K} - 1$ and left boundary as $k_l = k_c - \frac{1}{K}$ and right boundary as $k_r = k_c + \frac{1}{K}$. Then for each $k$, the probability is the mass from $k_l$ to $k_r$, *i.e.*, $\int_{k_l}^{k_r} \mathcal{N}(\mu_{h_c}, \sigma_{h_c}^2 \mathbf{I})$. And the first and last bins are curated by making sure the sum of the probability mass is 1. Then the *output distribution* is:

$$p_O(\mathbf{h}_c \mid \boldsymbol{\theta}^g, t; \phi) = \prod_{d=1}^{D} p_O^{(d)}\left(k\left(h_c^{(d)}\right) \mid \boldsymbol{\theta}^g, t; \phi\right), \tag{15}$$

where the function $k(\cdot)$ maps the variable to the corresponding bucket.

**Atom Types $\mathbf{h}_t$:** The atom types $\mathbf{h}_t$ are discrete variables with $K$ categories, where the corresponding parameter space lies in probability simplex thus the procedure is slightly different from the others. The *input distribution* for $\mathbf{h}_t$ is $p_I(\mathbf{h}_t \mid \boldsymbol{\theta}) = \prod_{d=1}^{D} \boldsymbol{\theta}^{h_t(d)}$, where $D$ is number if variables. And the input prior $\boldsymbol{\theta}_0^{h_t} = \frac{\mathbf{1}}{\mathbf{K}}$, where $\frac{\mathbf{1}}{\mathbf{K}}$ is the length $KD$ vector whose entries are all $\frac{1}{K}$. The *sender distribution*, could be derived with the central limit theorem, lies in the form of

$$p_S(\mathbf{y} \mid \mathbf{h}_t; \alpha) = \mathcal{N}\left(\mathbf{y} \mid \alpha\left(K\mathbf{e}_{\mathbf{h}_t} - \mathbf{1}\right), \alpha K\mathbf{I}\right) \tag{16}$$

where $\mathbf{1}$ is a vector of ones, $\mathbf{I}$ is the identity matrix, and $\mathbf{e}_j \in \mathbb{R}^K$ is a vector defined as the projection from the class index $j$ to a length $K$ one-hot vector (proof given by Graves et al. (2023)). In other words, each element of $\mathbf{e}_j$ is defined as $(\mathbf{e}_j)_k = \delta_{jk}$, where $\delta_{jk}$ is the Kronecker delta function. And $\mathbf{e}_{\mathbf{h}_t} \overset{\text{def}}{=} \left(\mathbf{e}_{\mathbf{h}_t^{(1)}}, \dots, \mathbf{e}_{\mathbf{h}_t^{(D)}}\right) \in \mathbb{R}^{KD}$.

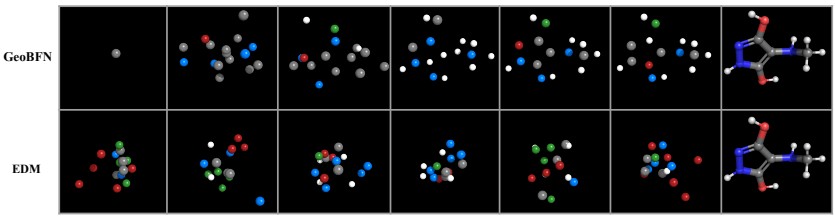

Figure 3: The Bayesian Flow and Diffusion Process of GeoBFN and EDM.

The *Bayesian update function* could be derived as $h\left(\boldsymbol{\theta}_{i-1}, \mathbf{y}, \alpha\right) = \frac{e^{\mathbf{y}}\boldsymbol{\theta}_{i-1}}{\sum_{k=1}^{K} e^{\mathbf{y}_k}(\boldsymbol{\theta}_{i-1})_k}$ (proof given by Graves et al. (2023)) . And similar to Eq. 13, the *Bayesian flow distribution* for $\mathbf{h}_t$ is as:

$$p_F(\boldsymbol{\theta}^{h_t} \mid \mathbf{h}_t; t) = \underset{\mathcal{N}\left(\mathbf{y}^{\mathbf{h_t}} \mid \beta(t)\left(K\mathbf{e}_{\mathbf{h}_t} - \mathbf{1}\right), \beta(t)K\mathbf{I}\right)}{\mathbb{E}} \delta(\boldsymbol{\theta}^{h_t} - \mathrm{softmax}(\mathbf{y}^{\mathbf{h_t}})) \tag{17}$$

With the network module $\Phi$, the *output distribution* could be obtained as

$$p_O^{(d)}(k \mid \boldsymbol{\theta}^g; t) = \left(\mathrm{softmax}\left(\Phi^{(d)}(\boldsymbol{\theta}^g, t)\right)\right)_k, p_O(\mathbf{h}_t \mid \boldsymbol{\theta}; t) = \prod_{d=1}^{D} p_O^{(d)}\left(h_t^{(d)} \mid \boldsymbol{\theta}^g; t\right) \tag{18}$$

**Training Objective**: By combining the different variables together, we could obtain the unified continuous-time loss for GeoBFN based on Eq. 25 to Eq. 41 in (Graves et al., 2023) as:

$$L^\infty(\mathbf{g}) = L^\infty(\langle \mathbf{x}, \mathbf{h}_c, \mathbf{h}_t \rangle) = \underset{t \sim U(0,1), p_F(\boldsymbol{\theta}^g | \mathbf{g}; t)}{\mathbb{E}} \left[ \frac{\alpha^g(t)}{2} \|\mathbf{g} - \Phi(\boldsymbol{\theta}^g, t)\|^2 \right]$$

$$= \underset{\substack{t \sim U(0,1), \\ \boldsymbol{\theta}^g \sim p_F(\cdot | \mathbf{g}; t)}}{\mathbb{E}} \left[ \frac{\alpha^x(t)}{2} \|\mathbf{x} - \Phi_{\mathbf{x}}\|^2 + \frac{\alpha^{\mathbf{h}_c}(t)}{2} \|\mathbf{h}_c - \Phi_{\mathbf{h}_c}\|^2 + \frac{\alpha^{\mathbf{h}_t}(t)}{2} \|\mathbf{h}_t - \Phi_{\mathbf{h}_t}\|^2 \right] \tag{19}$$

Where $\Phi_{\cdot}$ is short for $\Phi_{\cdot}(\boldsymbol{\theta}^g, t)$. The joint *Bayesian flow distribution* is decomposed as:

$$p_F(\boldsymbol{\theta}^g \mid \mathbf{g}; t) = p_F(\boldsymbol{\theta}^x \mid \mathbf{x}; t) p_F(\boldsymbol{\theta}^{h_c} \mid \mathbf{h}_c; t) p_F(\boldsymbol{\theta}^{h_t} \mid \mathbf{h}_t; t), \tag{20}$$

with $\alpha^x$, $\alpha^{h_c}$ and $\alpha^{h_t}$ refer to the corresponding accuracy scheduler (details provided by Graves et al. (2023)). And $\Phi_x$ is defined the same as in Eq. 14; while $\Phi_{h_c}$ is defined by the weighted average of different bucket centers with the output distribution in Eq. 15 as $\left(\sum_{k=1}^{K} p_O^{(1)}(k \mid \boldsymbol{\theta}, t) k_c, \dots, \sum_{k=1}^{K} p_O^{(D)}(k \mid \boldsymbol{\theta}, t) k_c\right)$; And for $\Phi_{h_t}$, it is defined as the $\sum_{k=1}^{K} p_O^{(d)}(k \mid \boldsymbol{\theta}; t) \mathbf{e}_k$ based on Eq. 18.

*Remark* 3.3. The GeoBFN defined in the above formulation satisfied the SE(3)-invariant condition in Theorem. 3.1.

**Sampling** GeoBFN will generate samples follow the graphical model in the recursive procedure as illustrated in Fig. 2a: *e.g.*, $g' \sim p_O(\cdot|\boldsymbol{\theta}_{i-1}) \rightarrow y \sim p_S(\cdot|g', \alpha) \rightarrow \boldsymbol{\theta}_i = h(\boldsymbol{\theta}_{i-1}, y, \alpha)$.

## 3.3 Overcome Noise Sensitivity in Molecule Geometry

One key obstacle of applying diffusion models to 3D molecule generation is the *noise sensitivity* property of the molecule geometry. The property of *noise sensitivity* seeks to state the fact: When noise is incorporated into the coordinates and displaces them significantly from their original positions, the bond distance between certain connected atoms may exceed the bond length range[1]. Under these circumstances, the point cloud could potentially lose the critical chemical information inherently encoded in the bonded structures; Another perspective stems from the reality that when noise is added to the coordinates, the relationships (distance) between different atoms could alter at a more rapid pace, e.g. modifying the coordinates of one atom results in altering its distance to all other atoms. Thus, the intermediate steps' structure in the generation procedure of diffusion models the intermediate steps' structure might be uninformative. And the majority of the information being acquired in the final few steps of generation (as depicted in Fig. 3).

A fundamental belief underpinning GeoBFN is that a smoother transformation during the generative process could result in a more favorable inductive bias according to (Graves et al., 2023). This process occurs within the parameter space of GeoBFN, which is regulated through the Bayesian update procedure. Specifically, samples exhibiting higher degrees of noise are assigned lesser weight during this update (refer to Eq. 11). This approach consequently leads to a significant reduction in variance within the parameter space as (Graves et al., 2023), which in turn facilitates the smooth transformation of molecular geometries. As illustrated in Fig. 3, this is evidenced by the gradual convergence of the structure of the intermediary steps towards the final structure, thus underscoring the effectiveness of smoother transformation.

### 3.4 OPTIMIZED DISCRETISED VARIABLE SAMPLING

Previous research (Hoogeboom et al., 2022; Xu et al., 2023; Wu et al., 2022) utilizes both the atom types $\mathbf{h}_t$ and charges $\mathbf{h}_c$ to represent the atomic properties. The $\mathbf{h}_c$ usually serves as an auxiliary loss for improving training which is not involved in determining the molecule graph during generation due to the insufficient modeling. However, there is redundant information between these two variables, since the $\mathbf{h}_t$ and $\mathbf{h}_c$ variables have a one-to-one mapping, *e.g.* the charge value 4 could be uniquely determined as the Carbon atom. We found that with advanced probabilistic modeling on discretized data, GeoBFN could conduct training and sampling only with $\mathbf{x}$ and $\mathbf{h}_c$. However, there exists a counterexample for the objective in Eq. 19 and the *output distribution* during sampling as in Eq. 15. As shown in Fig 5, the boundary condition for clamping the cumulative probability function in the bucket could cause the mismatch, *e.g.*, the true density should be centered in the center bucket while the *ouput distribution* instead put the most density in the first and last buckets which cause the mode-redundancy as shown in upper-left in Fig. 5. Though the weighted sum in Eq. 19 is optimized, the sampling procedure will rarely sample the center buckets. And such cases could be non-negligible in our scenarios, especially when the number of bins is small for low dimensional data. To alleviate this issue, we instead update the *output distribution* in the sampling procedure to:

$$\hat{\boldsymbol{k}}_c(\boldsymbol{\theta}, t) = \text{NEAREST\_CENTER}\left(\left[\sum_{k=1}^{K} p_O^{(1)}(k \mid \boldsymbol{\theta}, t)k_c, \ldots, \sum_{k=1}^{K} p_O^{(D)}(k \mid \boldsymbol{\theta}, t)k_c\right]\right) \qquad (21)$$

Function NEAREST\_CENTER compares inputs to the center bins $\vec{k}_c = \left(k_c^{(1)}, \ldots, k_c^{(D)}\right)$, and return the nearest center for each input value. The updated distribution is unbiased towards the training objective and also reduce the variance during generation which could be found in the trajectory of Fig.5.

## 4 EXPERIMENTS

### 4.1 EXPERIMENT SETUP

**Task and Datasets** We focus on the 3D molecule generation task following the setting of prior works (Gebauer et al., 2019; Luo & Ji, 2021; Satorras et al., 2021a; Hoogeboom et al., 2022; Wu et al., 2022). We consider both *Unconditional Molecular Generation* which assesses the capability to learn the underlying molecular data distribution and generate chemically valid and structurally diverse molecules and the *Conditional Molecule Generation* tasks which evaluate the capacity of generating molecules with desired properties. For *Conditional Molecule Generation*, we implement a conditional version GeoBFN with the details in the Appendix. The widely adapted **QM9** (Ramakrishnan et al., 2014) and the **GEOM-DRUG** (Gebauer et al., 2019; 2021) with large molecules are used for the experiments. And the data configurations directly follow previous work(Anderson et al., 2019; Hoogeboom et al., 2022; Xu et al., 2023)[2].

**Evaluation Metrics** The evaluation configuration follows the prior works (Hoogeboom et al., 2022; Wu et al., 2022; Xu et al., 2023). For the *Unconditional Molecular Generation*, the bond types are first predicted (single, double, triple, or none) based on pair-wise atomic distance and atom types in the 10000 generated molecular geometries (Hoogeboom et al., 2022). With the obtained molecular graph, we evaluate the quality by calculating both *atom stability* and *molecule stability* metrics. Besides, the *validity* (based on RDKIT) and *uniqueness* are also reported. Regarding the

---

[2]The official implementation is at https://github.com/AlgoMole/GeoBFN

Table 1: Results of atom stability, molecule stability, validity, validity×uniqueness (V×U), and novelty. A higher number indicates a better generation quality. The results marked with an asterisk were obtained from our own tests. And GeoBFN$_k$ denote the results of sampling the molecules with a specific number of steps $k$

| # Metrics | QM9 | | | | | DRUG | |
|---|---|---|---|---|---|---|---|
| | Atom Sta (%) | Mol Sta (%) | Valid (%) | V×U (%) | Novelty (%) | Atom Sta (%) | Valid (%) |
| Data | 99.0 | 95.2 | 97.7 | 97.7 | - | 86.5 | 99.9 |
| ENF | 85.0 | 4.9 | 40.2 | 39.4 | - | - | - |
| G-Schnet | 95.7 | 68.1 | 85.5 | 80.3 | - | - | - |
| GDM-AUG | 97.6 | 71.6 | 90.4 | 89.5 | 74.6 | 77.7 | 91.8 |
| EDM | 98.7 | 82.0 | 91.9 | 90.7 | 58.0 | 81.3 | 92.6 |
| EDM-Bridge | 98.8 | 84.6 | 92.0 | 90.7 | - | 82.4 | 92.8 |
| GEOLDM | 98.9 ± 0.1 | 89.4 ± 0.5 | 93.8 ± 0.4 | 92.7 ± 0.5 | 57.0 | 84.4 | **99.3** |
| **GEOBFN** $_{50}$ | 98.28 ± 0.1 | 85.11 ± 0.5 | 92.27 ± 0.4 | 90.72 ± 0.3 | 72.9 | 75.11 | 91.66 |
| **GEOBFN** $_{100}$ | 98.64 ± 0.1 | 87.21 ± 0.3 | 93.03 ± 0.3 | 91.53 ± 0.3 | 70.3 | 78.89 | 93.05 |
| **GEOBFN** $_{500}$ | 98.78 ± 0.8 | 88.42 ± 0.2 | 93.35 ± 0.2 | 91.78 ± 0.2 | 67.7 | 81.39 | 93.47 |
| **GEOBFN** $_{1k}$ | **99.08** ± 0.06 | **90.87** ± 0.2 | **95.31** ± 0.1 | **92.96** ± 0.1 | 66.4 | **85.60** | 92.08 |
| **GEOBFN** $_{2k}$ | **99.31** ± 0.03 | **93.32** ± 0.1 | **96.88** ± 0.1 | 92.41 ± 0.1 | 65.3 | **86.17** | 91.66 |

Table 2: Mean Absolute Error for molecular property prediction with 500 sampling steps. A lower number indicates a better controllable generation result.

| Property | $\alpha$ | $\Delta\varepsilon$ | $\varepsilon_{\text{HOMO}}$ | $\varepsilon_{\text{LUMO}}$ | $\mu$ | $C_v$ |
|---|---|---|---|---|---|---|
| Units | Bohr$^3$ | meV | meV | meV | D | $\frac{\text{cal}}{\text{mol}}$K |
| QM9* | 0.10 | 64 | 39 | 36 | 0.043 | 0.040 |
| Random* | 9.01 | 1470 | 645 | 1457 | 1.616 | 6.857 |
| $N_{\text{atoms}}$ | 3.86 | 866 | 426 | 813 | 1.053 | 1.971 |
| EDM | 2.76 | 655 | 356 | 584 | 1.111 | 1.101 |
| GEOLDM | 2.37 | 587 | 340 | 522 | 1.108 | 1.025 |
| **GEOBFN** | **2.34** | **577** | **328** | **516** | **0.998** | **0.949** |

Table 3: Ablation study, GeoBFN models molecule charge settings, the sampling step is set to 1,000.

| Charge Feature | Atom Stable (%) | Mol Stable (%) |
|---|---|---|
| discretised_basis | **99.08** | **90.87** |
| continuous_basis | 98.97 | 89.94 |
| discrete | 98.93 | 88.93 |
| discrete + continuous | 98.96 | 89.33 |
| discrete + discretised | 98.91 | 88.65 |

*Conditional Molecule Generation*, we evaluate our conditional version of GeoBFN on QM9 with 6 properties: polarizability $\alpha$, orbital energies $\varepsilon_{\text{HOMO}}$, $\varepsilon_{\text{LUMO}}$ and their gap $\Delta\varepsilon$, Dipole moment $\mu$, and heat capacity $C_v$. Following previous work Hoogeboom et al. (2022); Xu et al. (2023), the conditional GeoBFN is fed with a range of property $s$ to generate samples and the same pre-trained classifier $w$ is utilized to measure the property of generated molecule as $\hat{s}$. The *Mean Absolute Error* (MAE) between $s$ and $\hat{s}$ is calculated to measure whether the generated molecules is related to the conditioned property.

**Baselines** GeoBFN is compared with several advanced baselines including G-Schnet (Gebauer et al., 2019), Equivariant Normalizing Flows (ENF) (Satorras et al., 2021a) and Equivariant Graph Diffusion Models (EDM) with its non-equivariant variant (GDM) (Hoogeboom et al., 2022). Also with recent advancements, EDM-Bridge (Wu et al., 2022) which improves upon the performance of EDM by incorporating well-designed informative prior bridges and also GeoLDM (Xu et al., 2023) where a latent space diffusion model is applied are both included. To yield a fair comparison, all the method-agnostic configurations are set as the same. The implementation details could be found in Appendix. B.

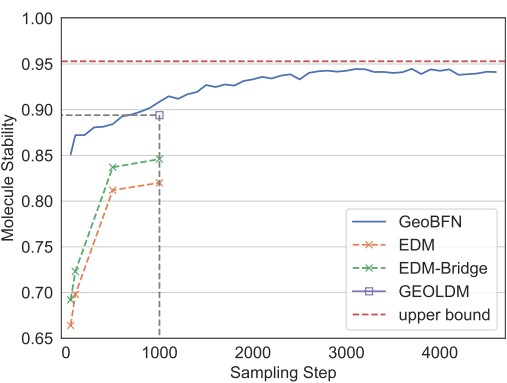

Figure 4: QM9 Molecule Stability wrt. Sampling Steps

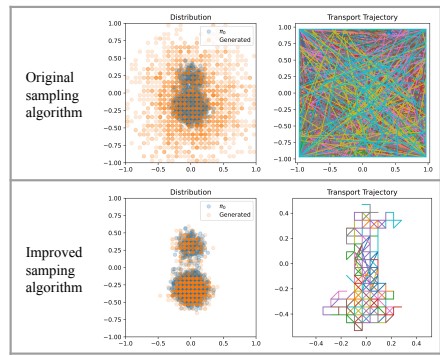

Figure 5: 2D Synthetic case of optimized synthetic example. In the left columns, generated samples are in orange, and data points are in blue.

## 4.2 MAIN RESULTS

The results of *Unconditional Molecular Generation* can be found in Tab. 1. We could observe that in both the **QM9** and **GEOM-DRUG** datasets, GeoBFN achieves a new state-of-the-art performance regarding both the quality and diversity of the generated molecules which demonstrates the huge potential of GeoBFN on geometry generative modeling. The phenomenon demonstrates that the GeoBFN does not hold the tendency to collapse to the subset of training data which could imply a probabilistic generalization ability and could be useful for several application scenarios; The *Conditional Molecule Generation* results can be found in Tab. 2. GeoBFN consistently outperforms other baseline models by an obvious margin in all conditional generation tasks. This clearly highlights the effectiveness and generalization capability of the proposed methods.

## 4.3 ANY-STEP SAMPLING

One notable property of GeoBFN is that training with the continuous-time loss, *e.g.*, Eq. 19, the sampling could be conducted with any steps without incurring additional training overhead. As shown in Tab. 1, GeoBFN could get superior performance compared to several advanced models with only 50 steps during sampling which brings $20\times$ speed-up during sampling due to the benefit of low variance parameter space. As we could find in Fig 4, with the sampling steps increasing from 50 to 4600, the molecule stability could be further boosted to approach the upper bound, *e.g.*, 94.25% with 4000 steps.

## 4.4 ABLATION STUDIES

We conduct ablation studies on the effect of input modalities in Tab. 3. We try different compositions and losses to represent the atom types, *discretised basis* refers to the case where the charge feature is used with discretised and the Gaussian basis, *i.e.*, $\phi_j(x) = \exp\left(-\frac{(x-\mu_j)^2}{2\sigma^2}\right)$ is used as functional embedding for charge; *continous basis* only differ in that the continous loss is utilized. The *discrete* refers to including the one-hot type representation; *discrete+continuous* refers to both the one-hot type and charge are included while continuous loss is included; Similar is the *discrete+continuous*. With only discretised variable utilized, the performance is superior to including the discrete variable which implies powerful probabilistic modeling capacity and the benefits of applying similar modality.

## 5 RELATED WORK

Previous molecule generation studies have primarily focused on generating molecules as 2D graphs (Jin et al., 2018; Liu et al., 2018; Shi et al., 2020), but there has been increasing interest in 3D molecule generation. With the increasing interest in 3D molecule generation, G-Schnet and G-SphereNet (Gebauer et al., 2019; Luo & Ji, 2021) respectively, employ autoregressive techniques to create molecules in a step-by-step manner by progressively connecting atoms or molecular fragments. These frameworks have also been extended to structure-based drug design (Li et al., 2021; Peng et al., 2022; Powers et al., 2022). There are approaches use atomic density grids that generate the entire molecule in a single step by producing a density over the voxelized 3D space (Masuda et al., 2020). Most recently, the attention has shifted towards using DMs for 3D molecule generation (Hoogeboom et al., 2022; Wu et al., 2022; Peng et al., 2023; Xu et al., 2023), with successful applications in target drug generation (Lin et al., 2022), antibody design (Luo et al., 2022), and protein design (Anand & Achim, 2022; Trippe et al., 2022). However, our method is based on the Bayesian Flow Network (Graves et al., 2023) objective and hence lies in a different model family which fundamentally differs from this line of research in both training and generation.

## 6 CONCLUSION

We introduce GeoBFN, a new generative framework for molecular geometry. GeoBFN operates in a differentiable parameter space for variables from different modalities. Also, the less variance in parameter space is naturally compatible with the *noise sensitivity* of molecule geometry. Given the appealing property, the GeoBFN achieves state-of-the-art performance on several 3D molecule generation benchmarks. Besides, GeoBFN can also conduct sampling with an arbitrary number of steps to reach an optimal trade-off between efficiency and quality (*e.g.*, $20\times$ speedup without sacrificing performance).

ACKNOWLEDGMENTS

The authors thank Yanru Qu for the helpful discussions and proofreading of the paper, as well as the anonymous reviewers for reviewing the draft. This work is supported by the National Science and Technology Major Project (2022ZD0117502), Natural Science Foundation of China (62376133) and Guoqiang Research Institute General Project, Tsinghua University (No. 2021GQG1012).

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

## A    EXPLANATION OF THE DATA EXCHANGE PERSPECTIVE OF BAYESIAN FLOW NETWORKS

In this section, we provide a brief overview of the Bayesian Flow Networks Graves et al. (2023) from a data exchange perspective. Bayesian Flow Networks (BFNs) is a new class of generative model that operates on the parameters of a set of independent distributions with Bayesian inference. The meta elements of BFNs are the *input distributions*, *sender distributions*, and *output distributions*. To start with, we denote the D-dimensional variable as $\mathbf{m} = \left(m^{(1)}, \ldots, m^{(D)}\right) \in \mathcal{M}^D$, and $\boldsymbol{\theta} = \left(\theta^{(1)}, \ldots, \theta^{(D)}\right)$ represent the parameters of a D-dimensional factorised distribution, *i.e.*, $p(\boldsymbol{m} \mid \boldsymbol{\theta}) = \prod_{d=1}^{D} p\left(m^{(d)} \mid \theta^{(d)}\right)$.

The basic logic of BFN could be better explained by the following communication example between the sender, referred to as Alice, and the receiver Bob. Alice aims to transfer some data to Bod in a progressive fashion, *i.e.*, at each timestep, Alice corrupts the data according to some channel noise, and then the noisy sample is transferred. The *sender distribution* is then defined to describe the noise-adding procedure, which is also a factorized distribution, $p_S(\mathbf{y} \mid \mathbf{m}; \alpha) = \prod_{d=1}^{D} p_S\left(y^{(d)} \mid m^{(d)}; \alpha\right)$. $\alpha$ refers to the accuracy parameter, $\alpha = 0$ refers to the information of the sample being totally destroyed by the noise, and with $\alpha$ increase the noisy sample will contain more information of the original sample. Intuitively, the sender distribution could be approximately understood as adding noise to each dimension of the data independently.

After receiving the noisy sample $\mathbf{y}$, Bob will first update an initial "guess" of what is the original sample behind the noisy sample, *i.e. input distribution*. Note that except for the noisy sample, Bob also knows the accuracy parameter $\alpha$ and noise formulation while not aware of the original sample $m$, *i.e.*, the noise level to create such a sample. The *input distribution* is initially a simple prior on the data space, *e.g.*, a standard Gaussian, lies in the mean-field family, $p_I(\mathbf{m} \mid \boldsymbol{\theta}) = \prod_{d=1}^{D} p_I\left(x^{(d)} \mid \theta^{(d)}\right)$. The parameter of input distribution will be updated through Bayesian inference, noted as $\boldsymbol{\theta}_i = h\left(\boldsymbol{\theta}_{i-1}, \mathbf{y}, \alpha_i\right)$. This update usually lies in a simple form, *e.g.* additive or weighted average.

After updating the parameter of *input distribution*, Bob has an "assistant" which will help to provide a better guess on the original data which generates the observed noisy sample. The assistant aims to exploit more context information between different dimensions, *e.g.*, the relationship between different pixels in an image, in contrast to updating each dimension independently as in the *input distribution*. Empirically, the assistant could be implemented by a neural network $\Psi$ which takes all parameters of input distribution for the prediction of parameters of each dimension, *i.e*, $\Psi(\theta) = \left(\Psi^{(1)}(\boldsymbol{\theta}, t), \ldots, \Psi^{(D)}(\boldsymbol{\theta}, t)\right)$.

The *output distribution* is then implied by the predicted parameter which lies in the formulation of $p_O(\mathbf{m} \mid \boldsymbol{\theta}, t) = \prod_{d=1}^{D} p_O\left(m^{(d)} \mid \Psi^{(d)}(\boldsymbol{\theta}, t)\right)$. Then Bob could construct a distribution to approximate the *sender distribution* at accuracy $\alpha$ by combining the *output distribution* with the known noise form, accuracy, *i.e.*, $p_R\left(\cdot \mid \boldsymbol{\theta}; t, \alpha\right) = \mathbb{E}_{p_O(\mathbf{x}' \mid \boldsymbol{\theta}; t)} p_S\left(\mathbf{y} \mid \mathbf{x}'; \alpha\right)$. Such distribution is called *receiver distribution*. The "assistant" of BFNs $\Psi$ is to minimize the KL divergence with a defined accuracy scheduler under different timesteps, *i.e.*, $D_{KL}\left(p_S\left(\cdot \mid \mathbf{m}; \alpha_i\right) \| p_R\left(\cdot \mid \boldsymbol{\theta}_{i-1}; t_{i-1}, \alpha_i\right)\right)$, which could also be interpreted as transmission cost under the bits-back coding scheme.

## B    IMPLEMENTATION DETAILS

The bayesian flow network is implemented with EGNNs Satorras et al. (2021b) by PyTorch (Paszke et al., 2017) package. We set the dimension of latent invariant features $k$ to 1 for QM9 and 2 for DRUG, which extremely reduces the atomic feature dimension. For the training of vector field network $v_\theta$: on QM9, we train EGNNs with 9 layers and 256 hidden features with a batch size 64; and on DRUG, we train EGNNs with 4 layers and 256 hidden features, with batch size 64. The model uses SiLU activations. We train all the modules until convergence. For all the experiments, we choose the Adam optimizer (Kingma & Ba, 2014) with a constant learning rate of $10^{-4}$ as our

default training configuration. The training on QM9 takes approximately 2000 epochs, and on DRUG takes 20 epochs.

## C    PROOF OF THEOREMS

In this Section, we provide the formal proof of the Theorem. 3.1 and Proposition. 3.2, as well as the detailed derivations for Equations.

### C.1    DISCUSSION ON THE TRANSLATIONAL INVARIANCE

*Remark* C.1. It is important to distinguish it from the rotation invariant. The rotational invariant is defined as $p(x) = p(\mathbf{R}x)$, while the translational is **not** as $p(x) = p(x + t)$ as such distribution can not integrate into one and hence does not exist. Fortunately, the freedom of translation could be eliminated by only focusing on learning distribution on the linear subspace where the center of gravity is always zero. This is, for all configurations on $\mathbb{R}^{n \times 3}$ space, the density on the zero CoM space is utilized to **represent** their density; It's important to note that the distribution is not defined for configurations outside the zero CoM space. However, it remains possible to leverage the distribution to provide a density-evaluation **(not probability density)** on the configurations outside the zero CoM space. This is achieved by projecting them back into the subspace. The evaluation procedure for configurations out of zero CoM space could only get a quantity defined artificially instead of the true density of some real distribution, e.g. It is referred to as "CoM-free density" in (Xu et al., 2022). Thus, there does not exist correctness issues.

#### C.1.1    ZERO CENTER OF MASS(COM) IN THE GEOBFN

Here we provide detailed discussions on optimizing the distribution in the zero CoM space. Recall the training objective in equation 8,

$$\mathcal{L}_{\text{VLB}}(\mathbf{x}) = \mathbb{E}_{p_\phi\left(\boldsymbol{\theta}_0^x, \ldots, \boldsymbol{\theta}_n^x\right)} \left[ \sum_{i=1}^{n} D_{KL}(p_S\left(\cdot \mid \mathbf{x}; \alpha_i\right) \| p_R(\cdot \mid \boldsymbol{\theta}_{i-1}^x; \alpha_i)) - \log p_\phi\left(\mathbf{x} \mid \boldsymbol{\theta}_n^x\right) \right], \quad (22)$$

where $p_\phi(\mathbf{x}|\boldsymbol{\theta}_n^x) = p_O(\mathbf{x}|\boldsymbol{\theta}_n^x, \phi)$. For learning a distribution on the zero CoM space of $\mathbb{R}^{n \times 3}$, the $p_S\left(\cdot \mid \mathbf{x}; \alpha_i\right)$, $p_R(\cdot \mid \boldsymbol{\theta}_{i-1}^x; \alpha_i)$ and $p_\phi\left(\mathbf{x} \mid \boldsymbol{\theta}_n^x\right)$ are all defined and supported on the zero CoM space, here $\sum_{i=1}^{n} \mathbf{x}_i = \mathbf{0}$ and $\sum_{i=1}^{n} \boldsymbol{\theta}^x = \mathbf{0}$. In other words, such distribution has no definition for variable $\mathbf{v} \in \mathbb{R}^{n \times 3}$ if $\sum_{i=1}^{n} \mathbf{v}_i \neq \mathbf{0}$. We then express the likelihood function of an isotropic diagonal Gaussian distribution, which is originally defined on the zero CoM space $((n - 1) \times 3$-dimensional), in the ambient space ($n \times 3$-dimensional) as (Hoogeboom et al., 2022):

$$\mathcal{N}_x\left(\boldsymbol{x} \mid \boldsymbol{\mu}, \sigma^2\mathbf{I}\right) = (\sqrt{2\pi}\sigma)^{-(n-1)\times 3} \exp\left(-\frac{1}{2\sigma^2}\|\boldsymbol{x} - \boldsymbol{\mu}\|^2\right) \quad (23)$$

Here $\sigma^2$ is the variance which is equivalent for each dimension. Recall the Eq. 35 in the (Graves et al., 2023), which shows that $D_{KL}(p_S\left(\cdot \mid \mathbf{x}; \alpha_i\right) \| p_R(\cdot \mid \boldsymbol{\theta}_{i-1}^x; \alpha_i))$ takes the form of $D_{KL}\left(\mathcal{N}\left(\mathbf{x}, \alpha_i^{-1}\boldsymbol{I}\right) \| \mathcal{N}(\Phi_{\mathbf{x}}(\boldsymbol{\theta}^g, t), \alpha_i^{-1}\boldsymbol{I})\right)$ which is the KL divergence between to diagonal Gaussian, then we derive the KL divergence for isotropic diagonal normal distributions of zero CoM with means represent on the ambient space. If $p_S = \mathcal{N}\left(\hat{\boldsymbol{\mu}}_1, \sigma^2\mathbf{I}\right)$ and $p_R = \mathcal{N}\left(\hat{\boldsymbol{\mu}}_2, \sigma^2\mathbf{I}\right)$ on subspace, where $\hat{\boldsymbol{\mu}}_1$ and $\hat{\boldsymbol{\mu}}_2$ is $(n - 1) \times 3$-dimension. Then the KL between them could be represented as:

$$D_{KL}(q\|p) = \frac{1}{2}\left[\frac{\|\hat{\boldsymbol{\mu}}_1 - \hat{\boldsymbol{\mu}}_2\|^2}{\sigma^2}\right] \quad (24)$$

There is an orthogonal transformation $Q$ which transforms the ambient space $\boldsymbol{\mu}_i \in \mathbb{R}^{n \times 3}$ where $\sum_i \boldsymbol{\mu}_i = \mathbf{0}$ to the subspace in the way that $\begin{bmatrix} \hat{\boldsymbol{\mu}} \\ \mathbf{0} \end{bmatrix} = \mathbf{Q}\boldsymbol{\mu}$. With $\|\hat{\boldsymbol{\mu}}\| = \|\begin{bmatrix} \boldsymbol{\mu} \\ \mathbf{0} \end{bmatrix}\| = \|\boldsymbol{\mu}\|$, there is $\|\hat{\boldsymbol{\mu}}_1 - \hat{\boldsymbol{\mu}}_2\|^2 = \|\boldsymbol{\mu}_1 - \boldsymbol{\mu}_2\|^2$. Hence we have:

$$D_{KL}\left(\mathcal{N}\left(\mathbf{x}, \alpha_i^{-1}\boldsymbol{I}\right) \| \mathcal{N}(\Phi_{\mathbf{x}}(\boldsymbol{\theta}^g, t), \alpha_i^{-1}\boldsymbol{I})\right) = \frac{\alpha_i}{2}\|\mathbf{x} - \Phi_{\mathbf{x}}\left(\boldsymbol{\theta}^g, t\right)\|^2 \quad (25)$$

Hence we demonstrate the correctness of our objective in Eq. 19.

### C.1.2 PROOF OF THE TRANSLATIONAL INVARIANT DENSITY EVALUATION PROCEDURE.

*Proof.* For an $n$-atom molecule $\mathbf{g} = \langle \mathbf{x}, \mathbf{h} \rangle$, the coordinate variable $\mathbf{x}$ has the dimension of $n \times 3$. Note that with the zero Center of Mass mapping (Xu et al., 2022; Satorras et al., 2021a), where we constrain the center of gravity as zero ($\sum_{i=1}^{n} \mathbf{x}_i = \mathbf{0}$), then variable $\mathbf{x}$ essentially lies in the $(n-1) \times 3$-dimensional linear subspace. The generative distributions $p_X$ mentioned in all of the related literature (Satorras et al., 2021a; Hoogeboom et al., 2022; Xu et al., 2022; 2023) is constrained in the zero Center of Mass space. This is, for samples in the ambient space, if $\sum_{i=1}^{n} \mathbf{x}_i \neq \mathbf{0}$, $p_X$ is not defined. The translational invariant property of distribution $p_X$ mentioned is not referred to the fact that $p_X(\mathbf{x}) = p_X(\mathbf{x} + \mathbf{t})$ for all translation vector $\mathbf{t}$ is satisfied in the ambient space with dimension $n \times 3$. Actually, such conditions could not be satisfied in any space (Satorras et al., 2021a). The translational invariant condition actually refers to the invariant function $f$ which could evaluate the density of all the ambient space based on $p_X$, the evaluated density by $f$ also referred to as "CoM-free standard density" in (Xu et al., 2022). The function $f$ is defined as

$$f(\mathbf{x}) = p_X(Q\mathbf{x}) \tag{26}$$

where $Q$ refers to the operation which maps the $\mathbf{x}$ to the zero Center of Mass space, *e.g.* in our work $Q$ is defined as

$$Q = I_3 \otimes \left( I_N - \frac{1}{N} \mathbf{1}_N \mathbf{1}_N^T \right), \quad \text{s.t.} \quad Q\mathbf{x} = \begin{bmatrix} \mathbf{x}_1 - \frac{\sum_{i=1}^{n} \mathbf{x}_i}{n} \\ \cdots \\ \mathbf{x}_n - \frac{\sum_{i=1}^{n} \mathbf{x}_i}{n} \end{bmatrix} \tag{27}$$

that subtracting mean $\frac{\sum_{i=1}^{n} \mathbf{x}_i}{n}$ from each $\mathbf{x}_i$, where $I_k$ denotes the $k \times k$ identity matrix and $\mathbf{1}_k$ denotes the k-dimensional vector filled with 1s. Then the density evaluation function $f$ is translational invariant in the ambient space:

$$f(\mathbf{x} + \mathbf{t}) = p_X(\hat{Q}(\mathbf{x} + \mathbf{t})) = p_X(\begin{bmatrix} \mathbf{x}_1 + \mathbf{t}_i - \frac{\sum_{i=1}^{n}(\mathbf{x}_i + \mathbf{t}_i)}{n} \\ \cdots \\ \mathbf{x}_n + \mathbf{t}_n - \frac{\sum_{i=1}^{n}(\mathbf{x}_i + \mathbf{t}_i)}{n} \end{bmatrix}) \tag{28}$$

Note $\mathbf{t}$ stands for a translation vector, which implies that $\mathbf{t}_1 = \cdots = \mathbf{t}_i = \mathbf{t}_n = \mathbf{C} \in \mathbb{R}^3$. Then we have:

$$f(\mathbf{x} + \mathbf{t}) = p_X(\begin{bmatrix} \mathbf{x}_1 + \mathbf{C} - \frac{\sum_{i=1}^{n}(\mathbf{x}_i + \mathbf{C})}{n} \\ \cdots \\ \mathbf{x}_n + \mathbf{C} - \frac{\sum_{i=1}^{n}(\mathbf{x}_i + \mathbf{C})}{n} \end{bmatrix}) = p_X(\begin{bmatrix} \mathbf{x}_1 - \frac{\sum_{i=1}^{n} \mathbf{x}_i}{n} \\ \cdots \\ \mathbf{x}_n - \frac{\sum_{i=1}^{n} \mathbf{x}_i}{n} \end{bmatrix}) = p_X(Q\mathbf{x}) = f(\mathbf{x}) \tag{29}$$

$\square$

Furthermore, the above proof has no constraint on the distribution $p_X$. This is, for any distribution on the zero Center of Mass space, the corresponding evaluation function defined in Eq. 26 is translational invariant.

### C.2 PROOF OF THEOREM. 3.1.

Given the above discussion on the translational invariance, for simplicity, we could only focus on the rotation transformation.

*Proof.* Recall the graphical model in Fig. 2, we could reformulate the density function in Eq. 7 as:

$$p_\phi(\mathbf{x}) = \int p_\phi(\mathbf{x} \mid \boldsymbol{\theta}_1^x, \cdots, \boldsymbol{\theta}_n^x) p_\phi(\boldsymbol{\theta}_1^x, \cdots, \boldsymbol{\theta}_n^x) d\boldsymbol{\theta}_{1:n}^x \quad \text{(definition of marginal)}$$

$$= \int p_\phi(\mathbf{x} \mid \boldsymbol{\theta}_n^x) p(\boldsymbol{\theta}_0) \prod_{i=1}^{n} p_U (\boldsymbol{\theta}_i \mid \boldsymbol{\theta}_{i-1}; \alpha_i) d\boldsymbol{\theta}_{1:n}^x. \tag{30}$$

Note that $p_\phi(\mathbf{x} \mid \boldsymbol{\theta}_n^x) = p_\phi(\mathbf{R}\mathbf{x} \mid \mathbf{R}\boldsymbol{\theta}_n^x) = p_O(\mathbf{R}(\mathbf{x}) \mid \mathbf{R}(\boldsymbol{\theta}_n^x); \phi)$ due to the property of EGNN, and $p(\boldsymbol{\theta}_0) = p(\mathbf{R}\boldsymbol{\theta}_0)$ since $\boldsymbol{\theta}_0 = \mathbf{0}$. Then we prove that $p_U (\boldsymbol{\theta}_i \mid \boldsymbol{\theta}_{i-1}; \alpha_i)$ satisfies the equivariant condition that $p_U (\boldsymbol{\theta}_i \mid \boldsymbol{\theta}_{i-1}; \alpha_i) = p_U (\mathbf{R}\boldsymbol{\theta}_i \mid \mathbf{R}\boldsymbol{\theta}_{i-1}; \alpha_i)$. Recall that $p_U (\boldsymbol{\theta}_i \mid \boldsymbol{\theta}_{i-1}; \alpha_i) =$

$\underset{p_O(\mathbf{y}_i|\boldsymbol{\theta}_{i-1};\alpha_i)}{\mathbb{E}} \delta\left(\boldsymbol{\theta}_i - h\left(\boldsymbol{\theta}_{i-1}, \mathbf{y}_i, \alpha_i\right)\right)$, then we have:

$$p_U\left(\mathbf{R}\boldsymbol{\theta}_i \mid \mathbf{R}\boldsymbol{\theta}_{i-1}; \alpha_i\right) = \underset{p_O(\mathbf{y}_i|\mathbf{R}\boldsymbol{\theta}_{i-1};\alpha_i)}{\mathbb{E}} \delta\left(\mathbf{R}\boldsymbol{\theta}_i - h\left(\mathbf{R}\boldsymbol{\theta}_{i-1}, \mathbf{y}_i, \alpha_i\right)\right)$$

$$= \int p_O\left(\mathbf{y}_i \mid \mathbf{R}\boldsymbol{\theta}_{i-1}; \alpha_i\right) \delta\left(\mathbf{R}\boldsymbol{\theta}_i - h\left(\mathbf{R}\boldsymbol{\theta}_{i-1}, \mathbf{y}_i, \alpha_i\right)\right) d\mathbf{y}_i \tag{31}$$

Then we apply integration-by-substitution and replace the variable $\mathbf{y}_i$ with a new variable $\mathbf{y}'_i$, *i.e.* $\mathbf{y}_i = \mathbf{R}\mathbf{y}'_i$, into the Eq. 31:

$$\int p_O\left(\mathbf{y}_i \mid \mathbf{R}\boldsymbol{\theta}_{i-1}; \alpha_i\right) \delta\left(\mathbf{R}\boldsymbol{\theta}_i - h\left(\mathbf{R}\boldsymbol{\theta}_{i-1}, \mathbf{y}_i, \alpha_i\right)\right) d\mathbf{y}_i$$

$$= \int p_O\left(\mathbf{R}\mathbf{y}'_i \mid \mathbf{R}\boldsymbol{\theta}_{i-1}; \alpha_i\right) \delta\left(\mathbf{R}\boldsymbol{\theta}_i - h\left(\mathbf{R}\boldsymbol{\theta}_{i-1}, \mathbf{R}\mathbf{y}'_i, \alpha_i\right)\right) d\mathbf{R}\mathbf{y}'_i$$

$$= \int p_O\left(\mathbf{R}\mathbf{y}'_i \mid \mathbf{R}\boldsymbol{\theta}_{i-1}; \alpha_i\right) \delta\left(\mathbf{R}\boldsymbol{\theta}_i - h\left(\mathbf{R}\boldsymbol{\theta}_{i-1}, \mathbf{R}\mathbf{y}'_i, \alpha_i\right)\right) |\det(\mathbf{R})| d\mathbf{y}'_i \tag{32}$$

The rotation matrix $\mathbf{R}$ is a SO(3) matrix, thus the $|\det(\mathbf{R})| = 1$. And for the continuous coordinate variable, the update function $h$ defined in Eq. 11 is also equivariant:

$$h\left(\mathbf{R}\boldsymbol{\theta}_{i-1}, \mathbf{R}\mathbf{y}_i, \alpha_i\right) = \frac{\mathbf{R}\boldsymbol{\theta}_{i-1}\rho_{i-1} + \mathbf{R}\mathbf{y}_i\alpha_i}{\rho_i} = \mathbf{R}h\left(\boldsymbol{\theta}_{i-1}, \mathbf{y}_i, \alpha_i\right) \tag{33}$$

Putting these conditions back to the Eq. 32, we have that

$$p_U\left(\mathbf{R}\boldsymbol{\theta}_i \mid \mathbf{R}\boldsymbol{\theta}_{i-1}; \alpha_i\right) = \int p_O\left(\mathbf{y}_i \mid \mathbf{R}\boldsymbol{\theta}_{i-1}; \alpha_i\right) \delta\left(\mathbf{R}\boldsymbol{\theta}_i - h\left(\mathbf{R}\boldsymbol{\theta}_{i-1}, \mathbf{y}_i, \alpha_i\right)\right) d\mathbf{y}_i$$

$$= \int p_O\left(\mathbf{R}\mathbf{y}'_i \mid \mathbf{R}\boldsymbol{\theta}_{i-1}; \alpha_i\right) \delta\left(\mathbf{R}\boldsymbol{\theta}_i - \mathbf{R}h\left(\boldsymbol{\theta}_{i-1}, \mathbf{y}'_i, \alpha_i\right)\right) |\det(\mathbf{R})| d\mathbf{y}'_i$$

$$= \int p_O\left(\mathbf{y}'_i \mid \boldsymbol{\theta}_{i-1}; \alpha_i\right) \delta\left(\boldsymbol{\theta}_i - h\left(\boldsymbol{\theta}_{i-1}, \mathbf{y}'_i, \alpha_i\right)\right) d\mathbf{y}'_i$$

$$= p_U\left(\boldsymbol{\theta}_i \mid \boldsymbol{\theta}_{i-1}; \alpha_i\right) \tag{34}$$

Hence the transitions on the $\boldsymbol{\theta}$ space are the Markov and equivariant to rotation as shown in Eq. 30. The initial state $\boldsymbol{\theta}_0$ is a zero vector $\mathbf{0}$ which is rotation invariant. To derive the rotation-invariant property of $p_\phi$, we will use the following Lemma, which is the direct application of Proposition 1 in (Xu et al., 2022). We changed the notation to make it consistent with our literature.

**Lemma C.2.** *(Xu et al., 2022) Let $p\left(\boldsymbol{\theta}_0\right)$ be an $SE(3)$-invariant density function, i.e., $p\left(\boldsymbol{\theta}_0\right) = p\left(T_g\left(\boldsymbol{\theta}_0\right)\right)$. If Markov transitions $p\left(\boldsymbol{\theta}_i \mid \boldsymbol{\theta}_{i-1}\right)$ are $SE(3)$-equivariant, i.e., $p\left(\boldsymbol{\theta}_i \mid \boldsymbol{\theta}_{i-1}\right) = p\left(T_g\left(\boldsymbol{\theta}_i\right) \mid T_g\left(\boldsymbol{\theta}_n\right)\right)$, then we have that the density $p\left(\boldsymbol{\theta}_n\right) = \int p\left(\boldsymbol{\theta}_0\right) p\left(\boldsymbol{\theta}_{1:n} \mid \boldsymbol{\theta}_0\right) d\boldsymbol{\theta}_{0:n}$ is also $SE(3)$-invariant. ($T_g$ stands for the SE(3) transformations.)*

For completeness, we also include the derivation of the lemma from (Xu et al., 2022):

$$p\left(T_g(\boldsymbol{\theta}_n)\right) = \int p\left(T_g(\boldsymbol{\theta}_0)\right) p\left(T_g(\boldsymbol{\theta}_{1:n}) \mid T_g(\boldsymbol{\theta}_0)\right) d\boldsymbol{\theta}_{0:n}$$

$$= \int p\left(T_g\left(\boldsymbol{\theta}_0\right)\right) \Pi_{i=1}^n p\left(T_g\left(\boldsymbol{\theta}_i\right) \mid T_g\left(\boldsymbol{\theta}_{i-1}\right)\right) d\boldsymbol{\theta}_{0:n}$$

$$= \int p\left(\boldsymbol{\theta}_0\right) \Pi_{i=1}^n p_\theta\left(T_g\left(\boldsymbol{\theta}_i\right) \mid T_g\left(\boldsymbol{\theta}_{i-1}\right)\right) d\boldsymbol{\theta}_{0:n} \quad (\text{ invariant prior } p\left(\boldsymbol{\theta}_0\right))$$

$$= \int p\left(\boldsymbol{\theta}_0\right) \Pi_{i=1}^n p_\theta\left(\boldsymbol{\theta}_i \mid \boldsymbol{\theta}_{i-1}\right) d\boldsymbol{\theta}_{0:n} \quad (\text{equivariant kernels } p\left(\boldsymbol{\theta}_i \mid \boldsymbol{\theta}_{i-1}\right))$$

$$= \int p\left(\boldsymbol{\theta}_0\right) p\left(\boldsymbol{\theta}_{1:n} \mid \boldsymbol{\theta}_0\right) d\boldsymbol{\theta}_{0:n}$$

$$= p\left(\boldsymbol{\theta}_n\right) \tag{35}$$

Given the invariant property of $\boldsymbol{\theta}_0$ and equivariant property of the transition $p_U\left(\boldsymbol{\theta}_i \mid \boldsymbol{\theta}_{i-1}; \alpha_i\right)$ and the $p_\phi(\mathbf{x} \mid \boldsymbol{\theta}_n^x)$, we could directly get the conclusion in Theorem. 3.1. Now we finish the proof. $\square$

### C.3 PROOF OF PROPOSITION. 3.2.

*Proof.* Then we derive the invariant property of the variational lower bound in of the variational lower bounds in equation 8:

$$\mathcal{L}_{VLB}(\mathbf{x}) = \mathop{\mathbb{E}}_{p_\phi(\boldsymbol{\theta}_0^x,\ldots,\boldsymbol{\theta}_n^x)} [\sum_{i=1}^n D_{KL}(p_S(\cdot \mid \mathbf{x};\alpha_i) \| p_R(\cdot \mid \boldsymbol{\theta}_{i-1}^x;\alpha_i)) - \log p_\phi(\mathbf{x} \mid \boldsymbol{\theta}_n^x)]$$

To start with, we consider the first term:

$$\mathop{\mathbb{E}}_{p_\phi(\boldsymbol{\theta}_0^x,\ldots,\boldsymbol{\theta}_n^x)} \sum_{i=1}^n D_{KL}(p_S(\cdot \mid \mathbf{x};\alpha_i) \| p_R(\cdot \mid \boldsymbol{\theta}_i^x;\alpha_i))$$

$$= \sum_{i=0}^{n-1} \mathop{\mathbb{E}}_{p_\phi(\boldsymbol{\theta}_i^x)} D_{KL}(p_S(\cdot \mid \mathbf{x};\alpha_i) \| p_R(\cdot \mid \boldsymbol{\theta}_i^x;\alpha_i)) \tag{36}$$

Note a natural conclusion from the Theorem. 3.1 is that the SE(3) invariance property is not only satisfied in the marginal distribution of the last time step variable $p(\boldsymbol{\theta}_n)$, but also for the distribution of any intermediate $p(\boldsymbol{\theta}_i)$. Such property could be justified based on the condition in Lemma. C.2. Actually, the proof of Theorem. 3.1 in the above section does not specify the time steps, hence the marginal distribution of any time step could be proved in exactly the same way. Consider the $i$-th term in the KL part of $\mathcal{L}_{\text{VLB}}(\mathbf{Rx})$:

$$\mathop{\mathbb{E}}_{p_\phi(\boldsymbol{\theta}_i^x)} D_{KL}(p_S(\cdot \mid \mathbf{Rx};\alpha_i) \| p_R(\cdot \mid \boldsymbol{\theta}_i^x;\alpha_i))$$

$$= \int p_\phi(\boldsymbol{\theta}_i^x) D_{KL}(p_S(\cdot \mid \mathbf{Rx};\alpha_i) \| p_R(\cdot \mid \boldsymbol{\theta}_i^x;\alpha_i)) d\boldsymbol{\theta}_i^x \tag{37}$$

we introduce the variable $\boldsymbol{\theta}_i'$ similar to Eq. 32, *i.e.* $\boldsymbol{\theta}_i = \mathbf{R}\boldsymbol{\theta}_i'$, and then we extend $i$-th term in the Eq. 36:

$$\int p_\phi\left(\mathbf{R}\boldsymbol{\theta}_i^{x'}\right) D_{KL}(p_S(\cdot \mid \mathbf{Rx};\alpha_i) \| p_R(\cdot \mid \mathbf{R}\boldsymbol{\theta}_i^{x'};\alpha_i)) |\det(\mathbf{R})| d\boldsymbol{\theta}_i^{x'} \tag{38}$$

As proved in the proof of Theorem. 3.1, $p_\phi$ is invariant and hence $p_\phi\left(\mathbf{R}\boldsymbol{\theta}_i^{x'}\right) = p_\phi(\boldsymbol{\theta}_i^{x'})$; Also the $|\det(\mathbf{R})| = 1$ for SO(3) rotation matrix. And then we discuss the KL divergence term:

$$D_{KL}(p_S(\cdot \mid \mathbf{Rx};\alpha_i) \| p_R(\cdot \mid \mathbf{R}\boldsymbol{\theta}_i^{x'};\alpha_i)) = \int p_S(\mathbf{y} \mid \mathbf{Rx};\alpha_i) \log \frac{p_S(\mathbf{y} \mid \mathbf{Rx};\alpha_i)}{p_R(\mathbf{y} \mid \mathbf{R}\boldsymbol{\theta}_i^{x'};\alpha_i)} d\mathbf{y}$$

$$= \int p_S(\mathbf{Ry'} \mid \mathbf{Rx};\alpha_i) \log \frac{p_S(\mathbf{Ry'} \mid \mathbf{Rx};\alpha_i)}{p_R(\mathbf{Ry'} \mid \mathbf{R}\boldsymbol{\theta}_i^{x'};\alpha_i)} \det(\mathbf{R})| d\mathbf{y'}$$

$$= \int p_S(\mathbf{y'} \mid \mathbf{x};\alpha_i) \log \frac{p_S(\mathbf{y'} \mid \mathbf{x};\alpha_i)}{p_R(\mathbf{y'} \mid \boldsymbol{\theta}_i^{x'};\alpha_i)} d\mathbf{y'} = D_{KL}(p_S(\cdot \mid \mathbf{x};\alpha_i) \| p_R(\cdot \mid \boldsymbol{\theta}_i^{x'};\alpha_i)) \tag{39}$$

Note that $p_S(\mathbf{y'} \mid \mathbf{x};\alpha_i) = p_S(\mathbf{Ry'} \mid \mathbf{Rx};\alpha_i)$ is due to that the sender distribution is isotropic; And for receiver distribution, the equivariant property that $p_R(\mathbf{y'} \mid \mathbf{x};\alpha_i) = p_R(\mathbf{Ry'} \mid \mathbf{Rx};\alpha_i)$ is guaranteed by both the parameterization of $p_O$ with Equivariant Graph Neural Network and the isotropic $p_S$. At last, we put the above conclusion back to Eq. 37, and we get that:

$$\mathop{\mathbb{E}}_{p_\phi(\boldsymbol{\theta}_i^x)} D_{KL}(p_S(\cdot \mid \mathbf{Rx};\alpha_i) \| p_R(\cdot \mid \boldsymbol{\theta}_i^x;\alpha_i))$$

$$= \int p_\phi(\boldsymbol{\theta}_i^x) D_{KL}(p_S(\cdot \mid \mathbf{Rx};\alpha_i) \| p_R(\cdot \mid \boldsymbol{\theta}_i^x;\alpha_i)) d\boldsymbol{\theta}_i^x$$

$$= \int p_\phi(\boldsymbol{\theta}_i^x) D_{KL}(p_S(\cdot \mid \mathbf{x};\alpha_i) \| p_R(\cdot \mid \boldsymbol{\theta}_i^x;\alpha_i)) d\boldsymbol{\theta}_i^x \tag{40}$$

$$= \mathop{\mathbb{E}}_{p_\phi(\boldsymbol{\theta}_i^x)} D_{KL}(p_S(\cdot \mid \mathbf{x};\alpha_i) \| p_R(\cdot \mid \boldsymbol{\theta}_i^x;\alpha_i))$$

And here we prove the first term in $\mathcal{L}_{VLB}(\mathbf{Rx})$ is equivalent to $\mathcal{L}_{VLB}(\mathbf{x})$. The second term could be derived in exactly the same way, and here we finish the proof. $\qquad\square$

### C.4 DERIVATION OF EQUATION 8

Note that the equation 8:

$$\mathcal{L}_{\text{VLB}}(\mathbf{x}) = \underset{p_\phi\left(\boldsymbol{\theta}_0^x, \dots, \boldsymbol{\theta}_n^x\right)}{\mathbb{E}} \left[ \sum_{i=1}^n D_{KL}(p_S\left(\cdot \mid \mathbf{x}; \alpha_i\right) \| p_R(\cdot \mid \boldsymbol{\theta}_{i-1}^x; \alpha_i)) - \log p_\phi\left(\mathbf{x} \mid \boldsymbol{\theta}_n^x\right) \right], \quad (41)$$

is the extension formulation of Eq. 1. To align the notation of Eq. 1 and equation 8, we use $\mathbf{x}$ in the following derivation. We first consider the term $-D_{KL}(q \| p_\phi\left(\mathbf{y}_1, \dots, \mathbf{y}_n\right))$ in Eq. 1, we put the Eq. 2

$$q = q\left(\mathbf{y}_1, \dots, \mathbf{y}_n \mid \mathbf{x}\right) = \prod_{i=1}^n p_S\left(\mathbf{y}_i \mid \mathbf{x}; \alpha_i\right) \quad (42)$$

And the $p_\phi\left(\mathbf{y}_1, \dots, \mathbf{y}_n\right))$ in Eq. 6 as

$$p_\phi\left(\mathbf{y}_1, \dots, \mathbf{y}_n\right) = \underset{p_\phi(\boldsymbol{\theta}_{0:n-1})}{\mathbb{E}} \left[ \prod_{i=1}^n \underset{p_O(\mathbf{x}_i' \mid \boldsymbol{\theta}_{i-1}; \phi)}{\mathbb{E}} \left[ p_S(\mathbf{y}_i \mid \mathbf{x}_i'; \alpha_i) \right] \right]$$

$$= \underset{p_\phi(\boldsymbol{\theta}_{0:n})}{\mathbb{E}} \prod_{i=1}^n p_R(\mathbf{y}_i \mid \boldsymbol{\theta}_{i-1}; \alpha_i) \quad (43)$$

Putting them together into the KL divergence term, and then we get the

$$D_{KL}(q \| p_\phi\left(\mathbf{y}_1, \dots, \mathbf{y}_n\right)) = \underset{\prod_{i=1}^n p_S(\mathbf{y}_i \mid \mathbf{x}; \alpha_i)}{\mathbb{E}} \log \frac{\prod_{i=1}^n p_S\left(\mathbf{y}_i \mid \mathbf{x}; \alpha_i\right)}{\underset{p_\phi(\boldsymbol{\theta}_{0:n-1})}{\mathbb{E}} \prod_{i=1}^n p_R(\mathbf{y}_i \mid \boldsymbol{\theta}_{i-1}; \alpha_i)}$$

$$= \underset{p_\phi(\boldsymbol{\theta}_{0:n-1}) \prod_{i=1}^n p_S(\mathbf{y}_i \mid \mathbf{x}; \alpha_i)}{\mathbb{E}} \log \frac{\prod_{i=1}^n p_S\left(\mathbf{y}_i \mid \mathbf{x}; \alpha_i\right)}{\prod_{i=1}^n p_R(\mathbf{y}_i \mid \boldsymbol{\theta}_{i-1}; \alpha_i)} \quad (44)$$

$$= \underset{p_\phi(\boldsymbol{\theta}_{0:n-1})}{\mathbb{E}} \sum_{i=1}^n D_{KL}(p_S\left(\cdot \mid \mathbf{g}; \alpha_i\right) \| p_R(\cdot \mid \boldsymbol{\theta}_{i-1}; \alpha_i))$$

And we have derived the first term in equation 8. And for the second term,

$$\log p_\phi(\mathbf{x} \mid \mathbf{y}_1, \cdots, \mathbf{y}_n) = \log p_\phi(\mathbf{x} \mid \boldsymbol{\theta}_0, \cdots, \boldsymbol{\theta}_n) \quad \text{(Graphical Model in Fig. 2)}$$

$$= \log p_\phi(\mathbf{x} \mid \boldsymbol{\theta}_n) \quad \text{(Markov Property of } \boldsymbol{\theta}) \quad (45)$$

And here we finish the derivation.

## D DETAILS ON CONDITIONAL GENERATION EXPERIMENTS

### D.1 PARAMETERIZATION AND SAMPLING

For the conditional experiments, we directly follow the conditional setting of previous literature (Hoogeboom et al., 2022). We discuss the details of the parameterization and sampling in the following. For conditional experiments, we add the property $\mathbf{c}$ as the extra input for the interdependency modeling module in Eq. 19. The conditional objective will be as:

$$L^\infty(\mathbf{g}, \mathbf{c})$$

$$= \underset{t \sim U(0,1), p_F(\boldsymbol{\theta}^g \mid \mathbf{g}; t)}{\mathbb{E}} \left[ \frac{\alpha^x(t)}{2} \|\mathbf{x} - \Phi_{\mathbf{x}}\|^2 + \frac{\alpha^{\mathbf{h}_c}(t)}{2} \|\mathbf{h}_c - \Phi_{\mathbf{h}_c}\|^2 + \frac{\alpha^{\mathbf{h}_t}(t)}{2} \|\mathbf{h}_t - \Phi_{\mathbf{h}_t}\|^2 \right] \quad (46)$$

Where $\Phi.(\boldsymbol{\theta}^g, t, \mathbf{c})$ is short for $\Phi.(\boldsymbol{\theta}^g, t, \mathbf{c})$.

For the sampling procedure, the property $\mathbf{c}$ and node number $M$ will be firstly sampled from a prior $p(\mathbf{c}, M)$ defined in (Hoogeboom et al., 2022). Here $p(\mathbf{c}, M)$ is computed on the training partition as a parametrized two-dimensional categorical distribution where the continuous variable $c$ is discretized into small uniformly distributed intervals. Then we could conduct generation as in Algorithm 3 based on the conditional output distribution $p_O(\cdot \mid \boldsymbol{\theta}, \mathbf{c}, t)$ base on $\Phi(\boldsymbol{\theta}, \mathbf{c}, t)$.

## D.2 Explanations on the Properties in Tab. 2

$\alpha$ Polarizability: Tendency of a molecule to acquire an electric dipole moment when subjected to anexternal electric field.

$\varepsilon_{\text{HOMO}}$: Highest occupied molecular orbital energy.

$\varepsilon_{\text{LUMO}}$: Lowest unoccupied molecular orbital energy.

$\Delta\varepsilon$ Gap: The energy difference between HOMO and LUMO.

$\mu$ : Dipole moment.

$C_v$ : Heat capacity at 298.15 K

## E    Detailed Algorithms for Training and Sampling

For a better understanding of the whole procedure in training and sampling, we involve the detailed algorithms and implements of functions in Algorithm 1, Algorithm 2 and Algorithm 3.

---

**Algorithm 1** Functions for GeoBFN

---

**function** DISCRETISED_CDF($\mu \in \mathbb{R}, \sigma \in \mathbb{R}^+, x \in \mathbb{R}$)

$\quad F(x) \leftarrow \frac{1}{2}\left[1 + \text{erf}\left(\frac{x-\mu}{\sigma\sqrt{2}}\right)\right]$

$\quad G(x) \leftarrow \begin{cases} 0 & \text{if } x \leq -1 \\ 1 & \text{if } x \geq 1 \\ F(x) & \text{otherwise} \end{cases}$

$\quad$ **Return** $G(x)$

**end function**

**function** OUTPUT_PREDICTION($\boldsymbol{\mu}_x \in \mathbb{R}^{D\times 3}, \boldsymbol{\mu}_h \in \mathbb{R}^D, t \in [0,1], \gamma_x, \gamma_h \in \mathbb{R}^+, t_{min} \in \mathbb{R}^+$)

$\quad$ # $t_{min}$ set to 0.0001 by default

$\quad$ **if** $t < t_{min}$ **then**

$\quad\quad \hat{\mathbf{x}}(\boldsymbol{\theta}, t) \leftarrow \mathbf{0}$

$\quad\quad \hat{\boldsymbol{\mu}}_h \leftarrow \mathbf{0}$

$\quad\quad \hat{\boldsymbol{\sigma}}_h \leftarrow \mathbf{1}$

$\quad$ **else**

$\quad\quad$ Input $(\boldsymbol{\mu}_x, \boldsymbol{\mu}_h, t)$ to network, receive $\hat{\boldsymbol{\epsilon}}(\boldsymbol{\theta}, t), \hat{\boldsymbol{\mu}}_h^\epsilon, \ln \hat{\boldsymbol{\sigma}}_h^\epsilon$ as output

$\quad\quad \hat{\mathbf{x}}(\boldsymbol{\theta}, t) \leftarrow \frac{\boldsymbol{\mu}_x}{\gamma_x} - \sqrt{\frac{1-\gamma_x}{\gamma_x}}\hat{\boldsymbol{\epsilon}}(\boldsymbol{\theta}, t)$

$\quad\quad \hat{\boldsymbol{\mu}}_h \leftarrow \frac{\hat{\boldsymbol{\mu}}_h}{\gamma_h} - \sqrt{\frac{1-\gamma_h}{\gamma_h}}\hat{\boldsymbol{\mu}}_h^\epsilon$

$\quad\quad \hat{\boldsymbol{\sigma}}_h \leftarrow \sqrt{\frac{1-\gamma_h}{\gamma_h}}\ln\hat{\boldsymbol{\sigma}}_h^\epsilon$

$\quad$ **end if**

$\quad$ **for** $d \in 1, \cdots, D, k \in K$ **do**

$\quad\quad \boldsymbol{p}_O^{(d)}(k \mid \boldsymbol{\theta}; t) \leftarrow$ DISCRETISED_CDF$(\hat{\mu}_h^{(d)}, \hat{\sigma}_h^{(d)}, k_r) -$ DISCRETISED_CDF$(\hat{\mu}_h^{(d)}, \hat{\sigma}_h^{(d)}, k_l)$

$\quad$ **end for**

$\quad$ **Return** $\hat{\boldsymbol{x}}(\boldsymbol{\theta}, t), \boldsymbol{p}_O(\cdot \mid \boldsymbol{\theta}; t)$

**end function**

---

---

**Algorithm 2** Training with continuous loss

---

**Require:** $\sigma_x, \sigma_h \in \mathbb{R}$, number of bins $K \in \mathbb{N}$
**Input:** coordinates $\boldsymbol{x} \in \mathbb{R}^{D \times 3}$, normalized charges $\boldsymbol{h} \in [\frac{1}{K} - 1, 1 - \frac{1}{K}]^D$
$t \sim U(0, 1)$
$\gamma_x \leftarrow 1 - \sigma_x^{2t}, \gamma_h \leftarrow 1 - \sigma_h^{2t}$
$\boldsymbol{\mu}_x \sim \mathcal{N}(\gamma_x, \gamma_x(1 - \gamma_x)\boldsymbol{I})$
$\boldsymbol{\mu}_h \sim \mathcal{N}(\gamma_h, \gamma_h(1 - \gamma_h)\boldsymbol{I})$
$\hat{\boldsymbol{x}}(\boldsymbol{\theta}, t), \boldsymbol{p}_O(\cdot \mid \boldsymbol{\theta}; t) \leftarrow \text{OUTPUT\_PREDICTION}(\boldsymbol{\mu}_x, \boldsymbol{\mu}_h, t, \gamma_x, \gamma_h)$
$\hat{\boldsymbol{k}}(\boldsymbol{\theta}, t) \leftarrow \left( \sum_k \boldsymbol{p}_O^{(1)} \boldsymbol{p}_O(k \mid \boldsymbol{\theta}; t) k_c, \ldots, \sum_k \boldsymbol{p}_O^{(D)}(k \mid \boldsymbol{\theta}; t) k_c \right)$
$L^\infty(\boldsymbol{x}) \leftarrow -\ln \sigma_x \sigma_x^{-2t} \|\boldsymbol{x} - \hat{\boldsymbol{x}}(\boldsymbol{\theta}, t)\|^2$
$L^\infty(\boldsymbol{h}) \leftarrow -\ln \sigma_h \sigma_h^{-2t} \left\| \boldsymbol{h} - \hat{\boldsymbol{k}}(\boldsymbol{\theta}, t) \right\|^2$
**Return** $L^\infty(\boldsymbol{x}) + L^\infty(\boldsymbol{h})$

---

**Algorithm 3** Sampling procedure

---

\# $\vec{k}_c = \left( k_c^{(1)}, \ldots, k_c^{(D)} \right)$
\# Function NEAREST\_CENTER compares inputs to the center bins $\vec{k}_c$,
\# and return the nearest center for each input value.
**Require:** $\sigma_x, \sigma_h \in \mathbb{R}^+$, number of steps $N \in \mathbb{N}$
$\boldsymbol{\mu}_x, \boldsymbol{\mu}_h \leftarrow \boldsymbol{0}$
$\rho_x, \rho_h \leftarrow 1$
**for** $i = 1$ to $N$ **do**
    $t \leftarrow \frac{i-1}{n}$
    $\gamma_x \leftarrow 1 - \sigma_x^{2t}, \gamma_h \leftarrow 1 - \sigma_h^{2t}$
    $\hat{\boldsymbol{x}}(\boldsymbol{\theta}, t), \boldsymbol{p}_O(\cdot \mid \boldsymbol{\theta}; t) \leftarrow \text{OUTPUT\_PREDICTION}(\boldsymbol{\mu}_x, \boldsymbol{\mu}_h, t, \gamma_x, \gamma_h)$
    $\alpha_x \leftarrow \sigma_x^{-2i/n} \left( 1 - \sigma_x^{2/n} \right)$
    $\alpha_h \leftarrow \sigma_h^{-2i/n} \left( 1 - \sigma_h^{2/n} \right)$
    $\hat{\boldsymbol{k}}_c(\boldsymbol{\theta}, t) \leftarrow \text{NEAREST\_CENTER}\left( \left[ \sum_k \boldsymbol{p}_O^{(1)}(k \mid \boldsymbol{\theta}; t) k_c, \ldots, \sum_k \boldsymbol{p}_O^{(D)}(k \mid \boldsymbol{\theta}; t) k_c \right] \right)$
    $\boldsymbol{y}_h \sim \mathcal{N}(\hat{\boldsymbol{k}}_c(\boldsymbol{\theta}, t), \alpha_h^{-1}\boldsymbol{I})$
    $\boldsymbol{y}_x \sim \mathcal{N}(\hat{\boldsymbol{x}}(\boldsymbol{\theta}, t), \alpha_x^{-1}\boldsymbol{I})$
    $\boldsymbol{\mu}_x, \boldsymbol{\mu}_h \leftarrow \frac{\rho_x \boldsymbol{\mu}_x + \alpha_x \boldsymbol{y}_x}{\rho_x + \alpha_x}, \frac{\rho_h \boldsymbol{\mu}_h + \alpha_h \boldsymbol{y}_h}{\rho_h + \alpha_h}$
    $\rho_x, \rho_h \leftarrow (\rho_x + \alpha_x), (\rho_h + \alpha_h)$
**end for**
$\hat{\boldsymbol{x}}(\boldsymbol{\theta}, 1), \boldsymbol{p}_O(\cdot \mid \boldsymbol{\theta}; 1) \leftarrow \text{OUTPUT\_PREDICTION}(\boldsymbol{\mu}_x, \boldsymbol{\mu}_h, 1, 1 - \sigma_x^2, 1 - \sigma_h^2)$
$\hat{\boldsymbol{k}}_c(\boldsymbol{\theta}, 1) \leftarrow \text{NEAREST\_CENTER}\left( \left[ \sum_k \boldsymbol{p}_O^{(1)}(k \mid \boldsymbol{\theta}; 1) k_c, \ldots, \sum_k \boldsymbol{p}_O^{(D)}(k \mid \boldsymbol{\theta}; 1) k_c \right] \right)$
**Return** $\hat{\boldsymbol{x}}(\boldsymbol{\theta}, 1), \hat{\boldsymbol{k}}_c(\boldsymbol{\theta}, 1)$

---

