# OpenReview forum: "Unified Generative Modeling of 3D Molecules with Bayesian Flow Networks"
_ICLR.cc/2024/Conference — ICLR 2024 oral_

### Official Review · Reviewer_iwqe · 2023-10-30

**Soundness:** 3 good
**Presentation:** 4 excellent
**Contribution:** 3 good
**Rating:** 8
**Confidence:** 4

**Summary:**

This paper proposes the use of newly formulated Bayesian Flow Networks for use in generating small molecules.

**Strengths:**

- The authors provide a very readable and accessible introduction to Bayesian Flow Networks.
- Well-formulated mathematical foundations for the proposed method.
- The authors compare to a large variety of benchmarks, showing improvement compared to all of them.
- The authors have run ablation studies that clarify the different design choices made.

**Weaknesses:**

- The noise-sensitivity section (3.3) is not very clear, the authors should describe in more detail the issue and why a variance-increasing versus variance-decreasing sampling procedure is an important design decision. The claim of "smoother information changes" especially seems intuitive yet subjective.
- The authors should explain how the objective is changing in equation 21, as it is not clear to me how this is improving the issue with sparser sampling of the center buckets.

**Questions:**

- Could you expand why larger numbers of generation steps seems to decrease novelty of the molecules?
- Is it fair to say that atom stability is easier (requires less generation steps) than molecular stability?
- Is it clear why the validity of the DRUG database samples seems to decrease with the number of generation steps, and is overall lower than some of the baselines? Is there something special about this dataset?

---

> ### Author Response · Authors · 2023-11-17
> **Response to Reviewer iwqe Part 1**
>
> Thanks a lot for the constructive comments and. recognition of our work. We address all your concerns in the following.
>
> **Q1. The noise-sensitivity section (3.3) is not very clear, the authors should describe in more detail the issue and why a variance-increasing versus variance-decreasing sampling procedure is an important design decision. The claim of "smoother information changes" especially seems intuitive yet subjective.**
>
> Sorry for causing the confusion.  Section 3.3 has been rewritten as suggested.
>
> - **Further clarification on noise sensitivity:**
>
> The property of noise sensitivity seeks to state the fact: When noise is incorporated into the coordinates and displaces them significantly from their original positions, the bond distance between certain connected atoms may exceed the bond length range[1]. Under these circumstances, the point cloud could potentially lose the critical chemical information inherently encoded in the bonded structures; Another perspective stems from the reality that when noise is added to the coordinates, the relationships (distance) between different atoms could alter at a more rapid pace, e.g. modifying the coordinates of one atom results in altering its distance to all other atoms.
>
> - **Towards the "smoother information changes" and  design decision:**
>
> We agree that such a claim could be subjective as the reviewer says and we have rephrased the presentation in the updated version to eliminate the misunderstanding. The opinion of "smoother information changes" is borrowed from the original BFN paper [2], which describes the generation process in the parameter space as regularized with the Bayesian update procedure. This is, noisier samples will be assigned with a smaller weight during the update (Eq. 11). Our objective here is to describe that during the diffusion process for generation, the intermediate steps' structure might be uninformative, with the majority of the information being acquired in the final few steps of generation (as depicted in Figure 3); On the other hand, as shown in Figure 3, the structure of the intermediate steps gradually converges to the final structure, suggesting a smoother increase in information. The key intuition is our belief that gradual changes could potentially provide a beneficial inductive bias for generation procedure design decisions as in [2]. The "entropy-increasing" and "entropy-decreasing"  are removed for a more strict and objective presentation.
>
> **Q2. The authors should explain how the objective is changing in equation 21, as it is not clear to me how this is improving the issue with the sparser sampling of the center buckets.**
>
> We apologize for any confusion caused. The original sampling process for the discretised variable of the BFN, as detailed in [2], is depicted in Algorithm 6. Here, the variable $k$ is sampled using the formula
>
> $$
> \mathbf{k} \sim \text{DISCRETISED\\_OUTPUT\\_DISTRIBUTION}\left(\boldsymbol{\mu}, t, 1-\sigma_1^{2 t}\right) =
> \text{DISCRETISED\\_CDF}\left(\mu_x^{(d)}, \sigma_x^{(d)}, k_r\right) - \text{DISCRETISED\\_CDF}\left(\mu_x^{(d)}, \sigma_x^{(d)}, k_l\right)$$
>
>
> In this equation, $\text{DISCRETISED\\_OUTPUT\\_DISTRIBUTION}$ denotes the probability of each bin and the sampling procedure essentially involves drawing a sample from a categorical distribution defined by $\text{DISCRETISED\\_OUTPUT\\_DISTRIBUTION}$.
>
> For instance, if the $\text{DISCRETISED\\_OUTPUT\\_DISTRIBUTION}$ is supported on {0,1,2} with corresponding probabilities {0.5,0,0.5}, the original sampling method would only draw from either {0} or {2}. However, in Eq. 21, we first compute the weighted average, which turns out to be $0.5\*0 + 0\*1 + 0.5\*2 = 1$, and subsequently select the nearest neighbor in the support set, which also yields the value $1$. It's important to note that this method aligns more closely with the training objectives articulated in Eq.19. We have revised the corresponding part as suggested.

---

> ### Author Response · Authors · 2023-11-17
> **Response to Reviewer iwqe Part 2**
>
> **Q3. The reason why larger numbers of generation steps seem to decrease the novelty of the molecules**
>
> This is a great question! Note that by increasing the discretization steps, the sampled distribution will approach the distribution implied by the fully continuous objective in Eq. (19). The novelty is defined as the proportion of generated samples that are unique and unseen in the training dataset. This reveals the consistency of model distribution in Eq. (19) and data distribution.
>
> Besides, the decrease in novelty could also raise a concern about an overfitting phenomenon. And we clarify the concerns in the following. As demonstrated in previous works [2,3], the concept of novelty must be examined alongside validity, uniqueness, and stability rates to get a comprehensive evaluation of the generative model. For a fair comparison with other models, e.g. EDM and GeoLDM, which use 1k steps to sample, we consider the performance of GeoBFN under the 1k steps. We could find that our method indeed demonstrates higher novelty compared to others. With the superior performance on the molecule quality metrics (stability, etc), our method also holds a higher novelty. This demonstrates that our proposed method shows better generalization ability and could better explore the molecule space instead of overfitting.
>
> **Q4. Relationship between atom stability and molecule stability**
>
> Yes, as the reviewer noticed, with fewer generation steps, the atom stability does not decrease obviously. Note that atom stability is defined as the proportion of atoms having the right valency, and molecular stability is defined as the proportion of generated molecules for which all atoms are stable [2]. Hence, molecule stability could be approximately seen as the $M$-th exponential to the atom stability where $M$ is the number of atom nodes.
>
> **Q5. Is it clear why the validity of the DRUG database samples seems to decrease with the number of generation steps, and is overall lower than some of the baselines? Is there something special about this dataset?**
>
> We borrow the discussion towards validity and stability in [2]. "Note that the validity is evaluated using RDkit. The molecule is built to contain only heavy atoms and RDKit will add hydrogens to each heavy atoms in such a way that the valency of each atom matches its atom type. Hence invalid molecules mostly appear when an atom has a valency bigger than expected. Experimentally,  it is observed that validity could artificially be increased by reducing the number of bonds. For example, predicting only single bonds was enough to obtain close to 100% of valid molecules on GEOM-DRUGS. On the contrary, the stability metrics directly model hydrogens and cannot be tricked as easily."
> As the reviewer noticed, the validity decreased with the number of generation steps and was even lower than some baselines on GEOM. This could reflect some intrinsic properties of GeoBFN. We hypothesize that the GeoBFN would tend to generate molecules with a more compact structure (as the generation variance is lower) and hence could get bigger valencies for atoms especially when the atom number is large. However, it is worth noticing that on the stability metrics which could not be tricked easily, GeoBFN still shows superior performance.
>
> -----
> [1] CRC Handbook.  2007.  "CRC Handbook of Chemistry and Physics", 88th Edition.
>
> [2] Emiel Hoogeboom, Victor Garcia Satorras, Clement Vignac, Max Welling 2022 "Equivariant Diffusion for Molecule Generation in 3D"
>
> [3] Minkai Xu, Alexander S. Powers, Ron O. Dror, Stefano Ermon, Jure Leskovec 2023 "Geometric Latent Diffusion Models for 3D Molecule Generation"

---

> > ### Comment · Reviewer_iwqe · 2023-11-18
> > **Thank you for your response**
> >
> > I would like to thank the authors for their in-depth responses to my questions. I will maintain the high score of 8 and hope that the paper is accepted.

---

### Official Review · Reviewer_mPXU · 2023-10-31

**Soundness:** 4 excellent
**Presentation:** 4 excellent
**Contribution:** 4 excellent
**Rating:** 8
**Confidence:** 2

**Summary:**

The authors apply Bayesian Flow networks (BFN) to the problem of modelling the 3D coordinates of molecules. The authors show how to make the BFN SE(3) equivariant and incorporate an SE(3) equivariant GNN into the BFN architecture for this. Strong performance is obtained on QM9 and GEOM-drugs. Additionally the model allows for flexible specification of the number of steps for sample generation - allowing trading off accuracy for speed.

**Strengths:**

- This paper is the first to apply BFN, a new class of generative models, to molecule generation.
- Good results: On both QM9 and GEOM-drug BFN outperform diffusion models which are a strong baseline.
- It is demonstrated that BFN achieve a better trade-off of sample quality vs sampling speed than diffusion models.
- The lower variance of the parameter space that BFN operate (relative to diffusion models which operate in sample space) seems advantageous - this is nicely visualised in Figure 3.

**Weaknesses:**

- I found the text inside the section "Overcome Noise Sensitivity In Molecule Geometry" unclear (see Questions below).

**Questions:**

In section 3.3 the text says
> Hence, GeoBFN implies an objective with smoother information changes.
What does "objective" refer to here? The text seems to be referring to sample generation (rather than training) but "objective" seems to imply a training objective?
Additionally the link between the marginals following an entropy increasing procedure implying an objective with smoother information changes is unclear to me - could the authors please elaborate?

---

> ### Author Response · Authors · 2023-11-17
> **Response to Reviewer mPXU**
>
> We thank the reviewer for the insightful comment and the recognition of our work. We address your concerns in the following:
>
> **Q1. Toward the section "Overcome Noise Sensitivity In Molecule Geometry":**
>
> -  What does "objective" refer to here?
>
> And yes, as the reviewer mentioned, here we are aiming to discuss the generation process. The "objective" should change to the "generation process".
>
> -  The link between the "entropy increasing procedure" and "smoother information changes"
>
> Sorry for causing the confusion.  The opinion of "smoother information changes" is borrowed from the original BFN paper [1], which describes the generation process in the parameter space as regularized with the Bayesian update procedure. This is, noisier samples will be assigned with a smaller weight during the update (Eq. 11). And we realized that the "entropy increasing procedure" is not strict enough considering the differentiable entropy of continuous variables.  Our objective here is to describe how during the diffusion process for generation, the intermediate steps' structure might be uninformative, with the majority of the information being acquired in the final few steps of generation (as depicted in Figure 3); On the other hand, as shown in Figure 3, the structure of the intermediate steps gradually converges to the final structure, suggesting a smoother increase in information.
> We have revised section 3.3 as suggested to prevent further misinterpretation.
>
> [1] Alex Graves, Rupesh Kumar Srivastava, Timothy Atkinson, Faustino Gomez 2023 "Bayesian Flow Networks"

---

> > ### Comment · Reviewer_mPXU · 2023-11-22
> >
> > Thank you for the clarification! I will stick to my score of 8 - I think this paper is good and recommend it for acceptance.

---

### Official Review · Reviewer_xteH · 2023-11-02

**Soundness:** 2 fair
**Presentation:** 2 fair
**Contribution:** 3 good
**Rating:** 8
**Confidence:** 4

**Summary:**

The authors apply a very recent generative modeling framework, [1], to the context of modeling biological molecules, and demonstrate improved performance on established molecular generation benchmarks.

------ Post-rebuttal ------

I believe my concerns about the theoretical presentation and clarity have been resolved. I believe this work is a solid contribution to the field, and I recommend its acceptance.

[1] Bayesian Flow Networks. https://arxiv.org/pdf/2308.07037.pdf

**Strengths:**

- The authors propose a novel model that conceptually simple to understand: the Bayes Flow Network introduced in [1] is applied to the context of molecular modeling by incorporating the equivariant structure of 3D molecular geometries.
- The empirical capabilities of the model, and its improved performance over existing works, is vetted with established baselines for molecular generation tasks.

**Weaknesses:**

-  There appears to be some significant issues with the formulation of the model. First, key equations are not derived or justified in the text or appendix. For example, how is the variational bound of the probabilistic model (Eq. 8) derived? Furthermore, how does it lead to Eq. 19? What is $L_\infty$? Is this the supremum norm? Why does minimizing this value lead to the correct parameters for the proposed model? It is difficult to verify the mathematical consistency of the model without these derivations. Second, the proof of Theorem 3.1 and Proposition 3.2 appear to be incomplete. For example, the proof of Theorem 3.1 ends mid-sentence. Moreover, I do not see anywhere a proof of translation invariance, only rotation invariance via the matrix $\mathbf{R}$. Additionally, how does Lemma C.1 establish Eq. 23? There appears to be major steps that are skipped in this proof.

- It is not entirely clear why Bayesian Flow Networks improve the performance of modelling 3D tasks (or perhaps specifically molecular generation tasks). The authors attempt to provide some intuition but I am not entirely convinced (see Questions).

Overall, I believe the work is very interesting but the manuscript requires some significant polish before it can be accepted.

**Questions:**

I could not find the definition of the subscripts $GeoBFN_{50}, GeoBFN_{100}, \dots, GeoBFN_{2k}$ in Table 1, Section 4. Where are these defined?

Can the authors clarify the meaning of this sentence in the last paragraph of Section 3.3: "The underlying reason lies in the fact that the marginal of θi in GeoBFN is in an entropy-increase procedure, e.g., from δ distribution(θ0) to the data distribution(θn). While in diffusion-based models, the marginal is in an entropy-decrease fashion, e.g., from a high-variance Gaussian distribution N (0, I ) to the data distribution." This sentence is very unclear to me. Additionally, the reasoning does not entirely connect for me. Isn't $\theta$ obtained from $\mathbf{y}$, which is an inherently noisy variable (i.e. Eq. 2)? So why do the authors claim that it has low entropy?

Are there any architectural / data preprocessing differences between the diffusion models (e.g. EDM) and the proposed GeoBFN? Are all improvements in performance attributable to the new training / sampling algorithm given by the Bayesian Flow Networks formulation [1]? Though the derivation is different, the training loss (Eq. 19) ultimately looks very similar to a diffusion model loss.

Can the authors provide a clear formulation of the training algorithm (e.g., via a latex algorithmic block) so it is more clear what is being calculated?

---

> ### Author Response · Authors · 2023-11-17
> **Response to Reviewer xteH Part 1**
>
> Thank you very much for the constructive suggestions and detailed comments. We address all your concerns in the following:
>
> **Q1. The presentation/formulation issues:**
>
> Thank you for bringing these issues to our attention, and we apologize for any confusion caused. We have addressed all of the mentioned issues in the following revised version.
>
> - **variational bounds in Eq. (8)**
>
> Actually, the variational bounds in Eq. (8) is exactly the detailed expansion of the variational bounds in Eq. (1), i.e.:
>
> $$
> \begin{align*}
> \log p_{\boldsymbol{\theta}}(\mathbf{g}) & \geq  \underset{\mathbf{y}_1, \ldots, \mathbf{y}_n \sim q}{\mathbb{E}}\left[\log  \frac{p\_\phi\left(\mathbf{g} \mid  \mathbf{y}_1, \ldots, \mathbf{y}_n\right) p\_\phi\left(\mathbf{y}_1, \ldots, \mathbf{y}_n\right)}{q\left(\mathbf{y}_1, \ldots, \mathbf{y}_n \mid  \mathbf{g}\right)}\right] \\newline
> & =-D\_{KL}\left(q \| p\_\phi\left(\mathbf{y}_1, \ldots, \mathbf{y}_n\right)\right)+\underset{\mathbf{y}_1, \ldots, \mathbf{y}_n \sim q}{\mathbb{E}} \log  \left[p\_\phi\left(\mathbf{g} \mid  \mathbf{y}_1, \ldots, \mathbf{y}_n\right)\right]
> \end{align*}
> $$
>
> In Section 2.2, we derive the formulation of each component: $q(\mathbf{y}_1, \ldots, \mathbf{y}_n)$ in Eq. (2); $p\_\phi\left(\mathbf{y}_1, \ldots, \mathbf{y}_n\right)$ in Eq. (6); The reconstruction term $p\_\phi\left(\mathbf{g} \mid  \mathbf{y}_1, \ldots, \mathbf{y}_n\right)$ is defined exactly as $p\_\phi\left(\mathbf{g} \mid  \theta_n  \right)$ in the framework. Push all this together into Eq. (1) we get the formulation of Eq. (8). And we include the key steps here.
>
> $$
> \begin{align*}
> D_{K L}\left(q\left(\boldsymbol{y}_1, \ldots, \boldsymbol{y}_n \mid\boldsymbol{x}\right) \| p\_\phi\left(\mathbf{y}_1, \ldots, \mathbf{y}_n\right)\right)&=\underset{\prod\_{i=1}^n p_S\left(\mathbf{y}_i \mid \mathbf{x} ; \alpha_i\right)}{\mathbb{E}} \log \frac{\prod\_{p\_\phi\left(\boldsymbol{\theta}\_{0: n-1}\right)}^n \prod\_{i=1}^n p_R\left(\mathbf{y}_i \mid \mathbf{x} ; \alpha_i\right)}{\mathbb{E}\left(\mathbf{y}_i \mid \boldsymbol{\theta}\_{i-1} ; \alpha_i\right)} \\newline
> & =\underset{p\_\phi\left(\boldsymbol{\theta}\_{0: n-1}\right)}{\mathbb{E}} \underset{\prod\_{i=1}^n}{\mathbb{E}} \mathbb{p _ { S } ( \mathbf { y } _ { i } | \mathbf { x } ; \alpha _ { i } )} \log \frac{\prod\_{i=1}^n p_S\left(\mathbf{y}_i \mid \mathbf{x} ; \alpha_i\right)}{\prod\_{i=1}^n p_R\left(\mathbf{y}_i \mid \boldsymbol{\theta}\_{i-1} ; \alpha_i\right)} \\newline
> & =\underset{p\_\phi\left(\boldsymbol{\theta}\_{0: n-1}\right)}{\mathbb{E}} \sum\_{i=1}^n D\_{KL}\left(p_S\left(\cdot \mid \mathbf{g} ; \alpha_i\right) \| p_R\left(\cdot \mid \boldsymbol{\theta}\_{i-1} ; \alpha_i\right)\right)
> \end{align*}
> $$
>
> A more formal and detailed derivation can be found in Appendix C.4 of the revised version
>
> - **derivation from Eq. (8) to Eq. (19) and explanation on $L_{\infty}$.**
>
> We apologize for any confusion caused. Our usage of the notation $L_{\infty}$ directly follows the notation in [1], which describes the objective function with an infinite number of continuous time steps. On the other hand, Equation (8) pertains to the objective function with a finite number of discrete time steps. The process of extending Equation (8) to account for continuous time steps can be found in [1] (specifically, from Equation (25) to Equation (41) for continuous variables). We added the reference as suggested to enhance the comprehensiveness of our paper.
>
> - **The correctness of the optimization objective $L_{\infty}$**
>
> As shown in the above discussion, the optimization objective $L_{\infty}$ can be viewed as an extension of the variational lower bound of the log-likelihood function to the case of infinite continuous time steps. Therefore, optimizing this objective essentially means maximizing the variational lower bounds of the log-likelihood function. Although there is a slight bias towards the maximum likelihood objective, such objectives are widely applied in training generative models, such as VAEs [1] and Diffusion Models [2]. We can observe the utilization of these objectives in practice.
>
> - **proof of Theorem 3.1 and Proposition 3.2 and the relationship between Lemma C.1 to Eq. (23)**
>
> Thanks a lot for pointing it out. The proof of Theorem 3.1 and Proposition 3.2 has been fully rewritten as suggested in the updated version in Appendix C.
>
> - **proof and discussion of translation invariance**
>
> The discussion and proof of translational invariance are added in Appendix C as the reviewer suggested. A more intuitive discussion can be found in the first part of our response to Reviewer m4nX's Q1.

---

> ### Author Response · Authors · 2023-11-17
> **Response to Reviewer xteH Part 2**
>
> **Q2. Definition of the subscripts: GeoBFN50, GeoBFN100,…, GeoBFN2k in Table 1, Section 4.**
>
> Apologies for the confusion caused. When training with a continuous time step objective, we have the flexibility to sample the molecule with arbitrary steps.
>
> In our approach, we use the term $\text{GeoBFN}\_{k}$ to denote the process of sampling the molecules with a specific number of steps. For instance, $\text{GeoBFN}\_{50}$ refers to the sampling of a molecule with 50 steps. To address your suggestion, we have included an explanation of this terminology in the updated draft.
>
> **Q3. Clarification of the last paragraph of Section 3.3.**
> - **Towards the "entropy" increase or "entropy" decrease:**
>
> Sorry for making the presentation unclear. After carefully checking the presentation in the mentioned sentence, the term "entropy" needs to be further clarified as the reviewer mentioned. Here we tend to distinguish the information changes between the BFNs and diffusion models. Entropy could be used as a good metric to analyze. For strict presentation, we need to limit the scoop of discussion on quantized space to guarantee the entropy is non-negative, e.g. images with a pixel value in {0,1,...,255}, instead of a continuous variable to avoid the discussion on differentiable entropy. In such cases, the entropy could be interpreted as the bits needed to describe/compress the distribution. Thus, in diffusion models, the sampling starts from a normal distribution, and with iterative refinement, the distribution approaches the data distribution which certainly has less entropy than the initial distribution("entropy" decrease); While in Bayesian Flow networks, the starting point of all sampling procedure is the same constant $\theta_0 = \textbf{0}$ where the entropy is zero. Here the sampled distribution will have entropy larger than the initial state("entropy" increase). The mentioned part is revised and the mentioned inaccurate parts are removed as suggested in the updated version to make it clear.
>
> Essentially, we would like to provide insight by distinguishing the source of randomness of the two different models. In the diffusion model, with recent advancements such as probability flow ode [3], the randomness of the system could be seen as obtained from the initial prior distribution. While in BFN, the randomness of the system is added gradually at each step by conducting a Bayesian update with the noisy variable.
>
> - **To the question of why the variable $\theta$ has lower entropy (variance):**
>
> As we only claim the modeling on the $\theta$ space enjoys lower "variance", we suppose the reviewer is concerned about the low-variance property. We take the modeling procedure of BFN on the continuous variable as an example. And yes, as the reviewer mentioned, $\theta$ is indeed obtained from $y$ which is the noisy variable. Recall the Bayesian update function in Eq. (11), i.e. $\boldsymbol{\mu}\_i = \frac{\boldsymbol{\mu}\_{i-1} \rho_{i-1}+\mathbf{y} \alpha}{\rho_i}$. The variance of $y$ is $\alpha^{-1}$ as defined in Eq. (10). Therefore, for $y$ with large variance, the $\alpha$ is small and hence contributes less to update $\theta$. In other words, the update of $\theta$ is dominated by those $y$ with small variance. Such a property of Bayesian update helps get a lower variance on $\theta$ compared to $y$. A visualizational analysis can be found in Figure 5 of [4]. The above discussion has been included as suggested.

---

> ### Author Response · Authors · 2023-11-17
> **Response to Reviewer xteH Part 3**
>
> **Q4. The architectural / data preprocessing differences to EDM and the attribution of the improvements of GeoBFN.**
>
> In our experiment, we ensured all components such as network architecture, data preprocessing, and evaluation pipeline were consistent with the settings in EDM to maintain fair comparisons.
> The performance enhancements were primarily due to the new training objective and sampling algorithm. As evident in Table 3's ablation studies, the performance was also improved with an optimized version of sampling.
> The training loss in Equation (19), as the reviewer mentioned, is similar to the diffusion objective. The distinguishing factors between BFNs and Diffusion models mainly lie in their differing modeling space and graphic models, as portrayed in Figure 2. The parameter space has the favorable characteristic of low variance, and the sampling process varies significantly from the diffusion model. In the latter, the final sample is forecasted and incorporated through a Bayesian update.  The joint modality modeling and previous properties could primarily contribute to the improved performance.
>
> **Q5. Clear formulation of the training algorithm.**
>
> Thanks for the suggestion. We have included the full training and sampling algorithm as suggested in Appendix E of the updated draft.
>
> ------
> [1] Diederik P Kingma, et. al. 2013. "Auto-Encoding Variational Bayes"
>
> [2] Jonathan Ho, et. al. 2020"Denoising Diffusion Probabilistic Models"
>
> [3] Yang Song, et. al. 2021. "Score-Based Generative Modeling through Stochastic Differential Equations"
>
> [4] Alex Graves, Rupesh Kumar Srivastava, Timothy Atkinson, Faustino Gomez 2023 "Bayesian Flow Networks"

---

> > ### Author Response · Authors · 2023-11-22
> > **Seeking feedbacks during the reviewer-author discussion period**
> >
> > Dear Reviewer xteH,
> >
> > Thank you for your insightful and detailed review comments and suggestions.
> >
> > This is a kind reminder that as the reviewer-author discussion period is ending soon, we look forward to hearing from you about your feedback on our response.
> >
> > In particular, we have polished the presentation and added more details as you suggested. We kindly invite you to review our point-by-point response which we have posted.
> >
> > Thank you again for your time and we sincerely look forward to your feedback!
> >
> > Best,
> >
> > Authors

---

### Official Review · Reviewer_m4nX · 2023-11-06

**Soundness:** 3 good
**Presentation:** 3 good
**Contribution:** 2 fair
**Rating:** 8
**Confidence:** 4

**Summary:**

Bayesian Flow Networks (BFN) are a recently proposed generative model, which uses diffusion in the inference process (like diffusion models), but for the generative process maintains a latent distribution parameters over the data, and updates these with Bayesian updates. These models have as advantage that they can handle discrete and discretized variables, besides continuous variables.

The authors propose to use BFNs to sample molecules, consisting of a collection of atoms, each with a continuous position and discrete atom type (or discretised atom charge). As the authors use a prior equivariant neural network, the resulting sampler is invariant to rotations (and to translations via centering). The authors show state-of-the-art performance in several unconditional and conditional sampling tasks.

**Strengths:**

- The method shows strong performance, exceeding prior methods.
- It's great to see a molecular sampling method used that handles the continuous positions and discrete atom types so naturally.
- The method improves consistently when more compute (=sampling steps) is used.

**Weaknesses:**

- I'm not so convinced about the translational equivariance of theorem 3.1. The concept "Zero of Mass" is not defined in the cited [1]. I suppose this is the space where $x$ has a zero center of mass. How does this affect $\theta$ and $y$? [2] gives a detailed analysis about how to handle translation invariance in diffusion, but it's not so clear to me how this applies immediately to a BFN. The proof of theorem 3.1 in the present manuscript says nothing about translations.
- The proposed method has limited novelty, as it combines a sampling method with an equivariant neural network to create an invariant sampler, as has been done many times previously, without other significant methodological innovation.

If the authors clear up the translational equivariance, I'll increase my score.

**Questions:**

- The boldness in the $V \times U$ and Novelty columns of table 1 appears incorrect.
- There appears to be an inconsistency in the definition of $p_U$ in Eq (5) of the manuscript and Eq (6) of [3]. Is the $y$ sampled from $p_O$ as the manuscript states, or from $p_S$ as [3] states?
- The authors write "For Conditional Molecule Generation, we implement a conditional version GeoBFN with the details in the Appendix", but I can't find this in the appendix.
- I'm quite surprised that sampling the charges via the discretized method outperforms sampling as discrete atom types. The atomic charge doesn't seems much like a continuum to me. Could the authors elaborate on this? Is it because the hydrogen / not hydrogen distinction is most important, which the discretized method is sensitive to?
- The results of EDM in table 1 seem worse than those reported in the EDM paper. Why is this?

Typo:
- Thm 3.1: transitional -> translational

Refs:
- [1] Köhler, Jonas, Leon Klein, and Frank Noé. 2020. “Equivariant Flows: Exact Likelihood Generative Learning for Symmetric Densities.”  http://proceedings.mlr.press/v119/kohler20a/kohler20a.pdf.
- [2] Xu, Minkai, Lantao Yu, Yang Song, Chence Shi, Stefano Ermon, and Jian Tang. 2021. “GeoDiff: A Geometric Diffusion Model for Molecular Conformation Generation,”https://openreview.net/forum?id=PzcvxEMzvQC.
- [3] Graves, Alex, Rupesh Kumar Srivastava, Timothy Atkinson, and Faustino Gomez. 2023. “Bayesian Flow Networks.” http://arxiv.org/abs/2308.07037.


----
Score raised following the discussion and the positive opinions of the other reviewers.

---

> ### Author Response · Authors · 2023-11-17
> **Response to Reviewer m4nX  Part 1**
>
> We thank the reviewer for the detailed comments and insightful suggestions. We address your concerns in the following:
>
>  **Q1. The translational equivariance of theorem 3.1.**
>
> Thank you for bringing this up, and we apologize for any confusion caused. The term "Zero of Mass" actually refers to a slightly abused concept of "Center-of-Mass Free" as discussed in Section 5 of [1], as well as "Zero Center of Mass" mentioned in [2]. We will now provide a thorough analysis of the translation invariance in Theorem 3.1, which is outlined below.
> The term SE(3)-invariant density is a concept that has been derived from previous research [2]. However, the notion of a "translational-invariant density" requires further clarification. In the normal Euclidean space, there is no distribution that can be truly translation-invariant, meaning that $p_X(x) = p_X(x+t)$  for all t, where t represents the translation vector. This condition implies that the probability density function is a constant function, which can not integrate into one [3].
>
> In existing literature, the term "translational-invariant property" actually refers to two key ideas.
> 1.  Firstly, it suggests that we can avoid modeling the freedom of translation of the geometries by learning a probability distribution only on the subspace where the center of mass is zero.
> 2. Secondly, it implies that it is possible to construct a function $f$ that can evaluate the density for samples in the original sample space, including samples with the center of mass is not zero.
>
> Intuitively, this function $f$ maps the samples to the zero center of mass space and returns the density of probability defined on that space. It is important to note that this function f does not correspond to the probability density function of any real distribution and is often referred to as the "CoM-free standard density" as described in [2]. Achieving translational invariance can be easily accomplished by constraining the sample space of the generative model to lie in the zero center of mass space. This can be achieved by moving the center of mass to zero for the generated samples, denoted as $x$ in our context.
>
> In Theorem 3.1, we have regularized all the parameters $\theta$, $y$, and $x$ to be centered around zero. Although this condition is sufficient to achieve translation invariance, it is not necessary as mentioned earlier. Specifically, regularizing the output of $\phi_x$ in the zero Center of Mass space should be sufficient to obtain this property. To ensure stable training, we also center all $\theta$ and $y$ around zero in the Center of Mass space.
>
> For more detailed and formal proof, as well as further discussion, please refer to Appendix C.1 of the updated version.
>
> **Q2. Towards limited novelty**
>
> Our paper is motivated by the natural compatibility of BFN[4] with the two key challenges in molecule generation. Apart from the  several innovations made for application on the specific task, we would like to highlight several contributions:
> - We present an alternative approach to understanding BFN through a simplified, graphical model summary of the original BFN paper outlined in [4]. This paper was based on a "communication" perspective that could often prove challenging to grasp and apply in other fields. The summarization could act as a beneficial resource aiding the comprehension of the complex properties and further exploration of this sophisticated generative model.
> - Another significant aspect is the demonstration of BFN compatibility with SE(3) invariant density modeling tasks, which is substantiated by the proof of Theorem 3.1. This realization is non-trivial given the fundamental differences between BNF and previous diffusion models.
>
> **Q3. The inconsistency of Eq. (5) in our paper and  Eq. (6) in [4]:**
>
> Thank you for pointing it out, and we apologize for any confusion caused. In our paper, we use Eq. (5) to describe the components of the generative process or the generative model. On the other hand, in [4], Eq. (6) is employed to formalize the training objective.
>
> In Eq. (5), the $p_O(y_i|\theta_{i-1};\alpha_i)$ should be changed to the so-called receiver distribution in [4], i.e.,
> $p_R (y_i|\theta_{i-1};\alpha_i) = \underset{p_O\left(\mathbf{x}^{\prime} \mid \boldsymbol{\theta}_{i-1}; \phi \right)}{\mathbb{E}} p_S\left(\mathbf{y} \mid \mathbf{x}^{\prime} ; \alpha_i\right)$.
>
> The typos have been fixed as suggested in the updated version.

---

> > ### Comment · Reviewer_m4nX · 2023-11-20
> > **Follow-up questions**
> >
> > I thank the authors for their response, revisions, and clarifications. I have some follow-up questions.
> >
> > - I understand that one can describe a generative process: Sample a non-centred position $\mathbb R^{n \times 3}$, then project to the zero CoM space by centring, so that you can define a $SE(3)$ equivariant sampling method. However, in your method, you are not just describing a sampling process, but also compute densities of molecular configurations. In order to compute the likelihood of a configuration in zero CoM space, naively, I would expect you'd need to integrate the distribution over all $\mathbb R^{n \times 3}$ configurations that differ only by CoM. I see no mention of this integration. Can you explain why this is not necessary and the likelihoods are correctly calculated?
> >
> > - In eq (8) of your revised version, you refer to $p_\phi(x|\theta^x_n)$. Where is this defined?
> >
> > - How does your alternative presentation differ from the VAE perspective in the BFN paper [4], eqns (17-20)?

---

> > > ### Author Response · Authors · 2023-11-20
> > > **Response to Follow-up Questions**
> > >
> > > Thanks a lot for your time and quick reply!
> > >
> > > **Q1. On the calculation of the densities:**
> > >
> > > This is a great question. For the translation invariance, though the community usually refers to the distribution/likelihood as the translation invariant[1,2,3], it is important to distinguish it from the rotation invariant. The rotational invariant is defined as $p(x)=p(\mathbf{R}x)$, while the translational is **not** as $p(x)=p(x+t)$ as such distribution can not integrate into one and hence does not exist.
> > >
> > > Fortunately, the freedom of translation could be eliminated by only focusing on learning distribution on the linear subspace where the center of gravity is always zero. This is, for all configurations on $\mathbb{R}^{n \times 3}$ space,  the density on the zero CoM space is utilized to **represent** their density[1,2];  It's important to note that the distribution is not defined for configurations outside the zero CoM space. However, it remains possible to leverage the distribution to provide a density-evaluation **(not probability density)** on the configurations outside the zero CoM space.  This is achieved by projecting them back into the subspace.
> > >
> > > The reviewer mentioned an alternative method to remove the freedom of translation by integrating the density over configurations that only differ by CoM across the entirety of the $\mathbb{R}^{n \times 3}$ space. However, this method could be intractable due to two main reasons.  Firstly, the continuous space of translation transformation for integration could be challenging. Secondly, if the density has a defined scope encompassing all the $\mathbb{R}^{n \times 3}$ configurations, the density would necessarily be impacted by the translation (can not be translational invariant) which is also undesirable.
> > >
> > > The evaluation procedure for configurations out of zero CoM space could only get a quantity defined artificially instead of the true density of some real distribution, e.g. it is referred to as "CoM-free density" in [1]. Thus,  there does not exist correctness issues. We understand that this part could easily lead to confusion. Therefore, we added more explanations as suggested to help clarify any misunderstandings.
> > >
> > > **Q2. The definition of $p_\phi(x|\theta_n^{x})$:**
> > >
> > > Sorry for causing the confusion. For the $p_\phi(\mathbf{x}|\theta_n)$, it is actually $
> > > p_O\left(\mathbf{x} \mid \boldsymbol{\theta}_n, \phi \right)
> > > $ as mentioned in  Page 3 after Eq. 6 in our paper.  We added the definition in Eq. 8 as suggested to eliminate the confusion.
> > >
> > > **Q3. How does your alternative presentation differ from the VAE perspective in the BFN paper [4], eqns (17-20)?**
> > >
> > > Our alternative graphical-model presentation mainly reveals the dependency between the variable $\boldsymbol {\theta}$ and the variable $\boldsymbol {y}$ and also helps to provide a clear comparison with diffusion models in Fig. 2. The VAE perspective in BFN paper [4], eqn3(17-20), shows the latent variable model formulation with only $\boldsymbol {y}$ as the latent variable while the network actually takes $\boldsymbol{\theta}$ as the input and output.  The proposed graphical presentation helps to figure out the Markov property on the $\boldsymbol{\theta}$ (while $\boldsymbol{y}$ is non-Markov), which is essential for understanding and proving the SE(3) invariant property of our model(GeoBFN).  We will carefully check the presentation as suggested to avoid overclaiming the contribution.
> > >
> > > If there is any further information you may need, please feel free to let us know!
> > >
> > > ---
> > > [1] Minkai Xu, Lantao Yu, Yang Song, Chence Shi, Stefano Ermon, and Jian Tang. 2021. “GeoDiff: A Geometric Diffusion Model for Molecular Conformation Generation,”
> > >
> > > [2] Emiel Hoogeboom, Victor Garcia Satorras, Clement Vignac, Max Welling 2022 "Equivariant Diffusion for Molecule Generation in 3D"
> > >
> > > [3]  Köhler, Jonas, Leon Klein, and Frank Noé. 2020. “Equivariant Flows: Exact Likelihood Generative Learning for Symmetric Densities.”

---

> ### Author Response · Authors · 2023-11-17
> **Response to Reviewer m4nX Part 2**
>
> **Q4. The details of the Conditional Molecule Generation experiment.**
>
> Thanks for bringing this to our attention. The generation of conditional molecules is directly performed after conducting conditional experiments in our previous work [5,6]. During conditional training, we utilize the interdependency modeling network $\phi$ in Eq.(19), which takes the property $c$ as an additional input, i.e., $\Phi(\theta,t,c)$.
>
> In the process of generating molecules conditionally, we first sample the property $c$ and the node number $M$ from a prior distribution $p(c,M)$ that is defined in [5]. The distribution $p(c,M)$ is computed on the training partition and is parameterized as a two-dimensional categorical distribution. The continuous variable $c$ is discretized into small, uniformly distributed intervals.
>
> For a more comprehensive explanation, please refer to the updated information in the Appendix D.
>
> **Q5. Discussion on sampling the charges via the discretized method outperforms sampling as discrete atom types.**
>
> This is a great question! Firstly, our main motivation for using charges for sampling is that we only need to model and sample on two similar modalities: discretized data type and continuous data type. This approach reduces the redundancy of information between the charges and types compared to previous literature [5, 6], which utilized all three modalities for modeling.
>
> Furthermore, the discretized charges allow for a more granular perspective of the distinctively varied electrostatic environment around different atoms. This introduces a more informed sampling framework that better integrates the chemical properties and reactivity.
>
> Additionally, as the reviewer mentioned, the atomic charge may not present a clear continuum, and there are nuances that justify the use of discretization. We fully agree with the reviewer's proposition that distinguishing between hydrogen and non-hydrogen atoms could be another important way to better understand the benefits of our proposed sampling method. Hydrogen is unique in terms of its proton count and chemical behavior, and while a categorical approach can differentiate between hydrogen and other atoms, a discretized approach can leverage this distinction even further. This could inspire a fruitful feature direction for further exploration.
>
> **Q6. The results of EDM in Table 1 seem worse than those reported in the EDM paper.**
>
> The results of EDM are directly copied from the EDM paper. After carefully checking the number, we make sure it is consistent with the number reported in their paper [5] and follow-ups [6].
>
> **Q7. Typos and boldness in Table 1.**
>
> Thanks again for the detailed comment. The mentioned issues are fixed in the updated version.
>
> [1] Köhler, Jonas, Leon Klein, and Frank Noé. 2020. “Equivariant Flows: Exact Likelihood Generative Learning for Symmetric Densities.”
>
> [2] Xu, Minkai, Lantao Yu, Yang Song, Chence Shi, Stefano Ermon, and Jian Tang. 2021. “GeoDiff: A Geometric Diffusion Model for Molecular Conformation Generation,”
>
> [3] Victor Garcia Satorras, Emiel Hoogeboom, Fabian B. Fuchs, Ingmar Posner, Max Welling. 2021. "E(n) Equivariant Normalizing Flows"
>
> [4] Alex Graves, Rupesh Kumar Srivastava, Timothy Atkinson, Faustino Gomez 2023 "Bayesian Flow Networks"
>
> [5] Emiel Hoogeboom, Victor Garcia Satorras, Clement Vignac, Max Welling 2022 "Equivariant Diffusion for Molecule Generation in 3D"
>
> [6] Minkai Xu, Alexander Powers, Ron Dror, Stefano Ermon, Jure Leskovec 2023 "Geometric Latent Diffusion Models for 3D Molecule Generation"

---

> ### Comment · Reviewer_m4nX · 2023-11-20
>
> I thank the authors for their follow-up response. My second and third question have been addressed. Regarding the first, I'm not sure I am fully convinced.
>
> I understand that you want to define a distribution on the space $\mathbb R^{3 N - 3}$, of the atomic positions modulo the centre of mass. My question is: what is the density on that space (which you use in the loss)?
>
> If I have an arbitrary density $p(x)$ on $\mathbb R^{3 N}$, and project it to a density $\hat p(\hat x)$ on $\hat x \in \mathbb R^{3 N - 3}$, then the value of the zero-CoM density is given by the integral:
> $$
> \hat p(\hat x) = \int_{\mathbb R^3}p(\hat x+t)dt
> $$
>
> This is not very practical and I understand you don't do this integration in practice. So how do you compute the densities in the zero CoM subspace?
>
> As an example, I understand $p_S(y | g, \alpha)$ to apply a Gaussian to $g$. If we assume $g$ to be zero CoM subspace, then  an isotropic Gaussian on $\mathbb R^{3 N}$ moves $y$ out of that subspace. So I suppose that your noise distribution is only supported in the zero CoM subspace? If so, does your likelihood include a factor $(2\pi)^{-(3N-3)/2}$? Similar for your distribution $p(g|\theta)$, which I suppose is a Gaussian with diagonal covariance matrix. Is that a diagonal $3N \times 3N$ matrix or a diagonal $(3N-3)\times (3N-3)$ matrix? If it's the former, I suppose that it doesn't project to a diagonal $(3N-3)\times (3N-3)$ matrix. Do you then explicitly compute the determinant of that matrix to find the likelihood?
>
> Prior works have worked with zero CoM variables in a diffusion context, where I understand from [1] that things simplify, but I haven't seen a convincing argument why the same applies for the BFN.
>
> [1] Minkai Xu, Lantao Yu, Yang Song, Chence Shi, Stefano Ermon, and Jian Tang. 2021. “GeoDiff: A Geometric Diffusion Model for Molecular Conformation Generation,”

---

> > ### Author Response · Authors · 2023-11-20
> > **Thanks for your reply！**
> >
> > The response is based on Appendix A in EDM (https://arxiv.org/pdf/2203.17003.pdf)[1]. We strongly encourage its use as a reference point for a comprehensive understanding.
> >
> > Firstly, we would like to clarify that $p_S(y|g,\alpha)$  and also $p_R(y|\theta,\alpha)$ is indeed defined and supported in the **zero Center of Mass space** as you supposed.
> >
> > And yes, consider the Euclidean variable $\boldsymbol{x} \in \mathbb{R}^{N \times 3}$ in linear subspace  $\sum_{i} \boldsymbol{x}\_{i} = \boldsymbol{0} $  (Zero Center of Mass space). We could place a normal distribution on the subspace $\mathbb{R}^{(N-1) \times 3}$, while the likelihood could be expressed as [1]:
> > $$\mathcal{N}\_{x}(\boldsymbol{x} \mid \boldsymbol{\mu}, \sigma\^{2} \boldsymbol{I})=(\sqrt{2 \pi} \sigma)^{-(N-1) \cdot 3} \exp \left(-\frac{1}{2 \sigma^2}\\|\boldsymbol{x}-\boldsymbol{\mu}\\|^2\right)$$
> > Note $\boldsymbol{\mu}$  is in the same subspace with  $\boldsymbol{x}$. (Dimensions of  $\boldsymbol{\mu}$   and $\boldsymbol{x}$ are all $ N \times 3 $).  As in the subspace, the corresponding diagonal covariance matrix is $(3N-3) \times (3N-3)$. And we do not need to compute the determinant to get the likelihood.  Also the constant $(\sqrt{2 \pi} \sigma)^{-(N-1) \cdot 3}$ is ignored during training.
> >
> > For our training objective, as shown in Eq. 8, we need to calculate the term $\sum_{i=1}^n D_{K L}\left(p_S\left(\cdot \mid \mathbf{x} ; \alpha_i\right) \| p_R\left(\cdot \mid \boldsymbol{\theta}_{i-1}^x ; \alpha_i\right)\right)$, this is the KL divergence between two diagonal normal distributions on the subspace which is exactly similar to the $\mathcal{L}_t=- D\_{K L}\left(q\left(\boldsymbol{z}_s \mid \boldsymbol{x}, \boldsymbol{z}_t\right) \| p\left(\boldsymbol{z}_s \mid \boldsymbol{z}_t\right)\right)$ (Eq. 17) of  EDM [1] ( Also KL between two subspace normal distribution).
> >
> > Note the variance $\sigma$ of $p_S$ and $p_R$  are the same for each time step. If $p_S=\mathcal{N}\left(\hat{\boldsymbol{\mu}\_{1}}, \sigma^2 \mathbf{I}\right)$ and $p_R = \mathcal{N}\left(\hat{\boldsymbol{\mu}\_{2}}, \sigma^2 \mathbf{I}\right)$ on subspace, where $\hat{\boldsymbol{\mu}\_{1}}$ and $\hat{\boldsymbol{\mu}\_{2}}$  is  $(N-1) \times 3$-dimension. And the KL between them is $\mathrm{KL}(p_S \\| p_R)=\frac{1}{2}\left[\frac{\left\|\hat{\boldsymbol{\mu}_1}-\hat{\boldsymbol{\mu}_2}\right\|^2}{\sigma^2}\right]$.
> > We could have an **orthogonal** transformation $Q$ which transforms the ambient space $\boldsymbol{\mu}\_{i} \in \mathbb{R}^{N \times 3}$ where  $\sum\_{i} \boldsymbol{\mu}\_{i}=\mathbf{0}$ to the subspace in the way that
> >
> > $\left[\begin{array}{l}
> > \hat{\boldsymbol{\mu}} \\\\
> > \mathbf{0}
> > \end{array}\right]=\mathbf{Q} \boldsymbol{\mu}$. With $\\|\hat{\boldsymbol{\mu}}\\|=\left\\|\left[\begin{array}{c}
> > \tilde{\boldsymbol{\mu}} \\\\
> > \mathbf{0}
> > \end{array}\right]\right\|= \|\boldsymbol{\mu}\\|$, there is $\left\\|\hat{\boldsymbol{\mu}}_1-\hat{\boldsymbol{\mu}}_2\right\\|^2=\left\\|\boldsymbol{\mu}_1-\boldsymbol{\mu}_2\right\\|^2$. And this justified the correctness of our final objective in Eq. 19 which is calculated in the ambient space with dimension $ N \times 3 $.
> >
> > ---
> > [1]. Emiel Hoogeboom, Victor Garcia Satorras, Clement Vignac, Max Welling 2022 "Equivariant Diffusion for Molecule Generation in 3D"

---

> > > ### Author Response · Authors · 2023-11-22
> > > **Thanks a lot for your time！**
> > >
> > > Dear Reviewer m4nX,
> > >
> > > We are deeply appreciative of your constructive review comments and active engagement in the discussions.  As the reviewer-author discussion period is ending soon, if there are still any concerns/questions about the translational invariance, please don't hesitate to let us know!
> > >
> > > Thank you again for your time and we sincerely look forward to your feedback!
> > >
> > > Best,
> > >
> > > Authors

---

> > > > ### Comment · Reviewer_m4nX · 2023-11-22
> > > >
> > > > Dear authors,
> > > >
> > > > Thank you for your response. If you're using isotropic Gaussians everywhere, I am inclined to believe that a similar reasoning as [1] for diffusion models applies here and believe that all likelihoods are computed correctly, perhaps up to irrelevant constants. However, if you're using Gaussians that are non-isotropic (even if they are diagonal, so a diagonal matrix with differing values on the diagonal), the connection to diffusion models becomes much less clear and I'd need to see a thorough proof that you're computing the loss correctly.
> > > >
> > > > Can you confirm that you're using isotropic Gaussians throughout?
> > > >
> > > > If so, this concern has been addressed, though I would very much appreciate a more thorough discussion/proof in the final version of your paper why your centring approach is valid in the BFN context.

---

> > > > > ### Author Response · Authors · 2023-11-22
> > > > > **Thanks for your comment!**
> > > > >
> > > > > Dear Reviewer m4nX,
> > > > >
> > > > > Thanks a lot for your time and response!
> > > > > As you mentioned, we confirm that we are using the isotropic Gaussian distribution. This directly follows the original BFN [1] where the sender distributions for continuous variable is also isotropic Gaussian.
> > > > >
> > > > > Besides, we have added a more thorough discussion in Appendix C.1.1 as you suggested. We deeply appreciate our conversations with you, as they have profoundly helped us in developing more rigorous and robust representations.
> > > > >
> > > > > We hope that we have addressed all your concerns. If there is any further information you may need, please feel free to let us know!
> > > > >
> > > > > Thank you again for your valuable time and we sincerely look forward to your feedback!
> > > > >
> > > > > Best,
> > > > >
> > > > > Authors
> > > > >
> > > > > ----
> > > > > [1] Alex Graves, Rupesh Kumar Srivastava, Timothy Atkinson, Faustino Gomez 2023 "Bayesian Flow Networks"

---

> > > > > > ### Comment · Reviewer_m4nX · 2023-11-22
> > > > > >
> > > > > > Thanks for your response. My concerns have been satisfactorily addressed and I have raised my score.

---

### Author Response · Authors · 2023-11-17
**General Response**

We are deeply appreciative of all the reviewers for their insightful comments and valuable suggestions!

We have responded to each question individually and, in accordance with your suggestions, have revised the paper. This includes **further information about the implementation**,  **detailed descriptions of the algorithms**, and **comprehensive proofs**. Besides, the presentation issues mentioned have been efficiently resolved by revising the statement in a more rigorous way.

Due to the page limit, most of them are updated in the Appendix. The updated text is colored in blue.

We hope that our response, along with the revised paper, could address your concerns!

---

### Meta-Review · Area_Chair_8Gaf · 2023-12-07

**Metareview:**

The paper introduces Geometric Bayesian Flow Networks for modelling molecules in an SE-(3) invariant way. The model achieves SOTA on common benchmarks and can be up to 20x faster in sampling molecules.

**Justification For Why Not Higher Score:**

NA

**Justification For Why Not Lower Score:**

cutting edge methodology for a highly active topic, achieving benchmark SOTA while being fast; consistent accept votes (8) of reviewers.

---

### Decision · Program_Chairs · 2024-01-16

Accept (oral)